# Deep learning-based incoherent holographic camera enabling acquisition of real-world holograms for holographic streaming system

Hyeonseung Yu [1,5], Youngrok Kim [2,5], Daeho Yang [1,4], Wontaek Seo[1], Yunhee Kim[1], Jong-Young Hong[1], Hoon Song[1], Geeyoung Sung[1], Younghun Sung[1], Sung-Wook Min [2] ✉ & Hong-Seok Lee [3] ✉

While recent research has shown that holographic displays can represent photorealistic 3D holograms in real time, the difficulty in acquiring high-quality real-world holograms has limited the realization of holographic streaming systems. Incoherent holographic cameras, which record holograms under daylight conditions, are suitable candidates for real-world acquisition, as they prevent the safety issues associated with the use of lasers; however, these cameras are hindered by severe noise due to the optical imperfections of such systems. In this work, we develop a deep learning-based incoherent holographic camera system that can deliver visually enhanced holograms in real time. A neural network filters the noise in the captured holograms, maintaining a complex-valued hologram format throughout the whole process. Enabled by the computational efficiency of the proposed filtering strategy, we demonstrate a holographic streaming system integrating a holographic camera and holographic display, with the aim of developing the ultimate holographic ecosystem of the future.

For several decades, holographic displays have been considered primary candidates for future 3D displays, as they provide natural viewing experiences that support physically accurate 3D cues, including accommodation cues[1]. Moreover, holographic displays can be realized in both slim-panel displays and augmented reality (AR) and virtual reality (VR) near-eye displays[2,3]. Low image quality and narrow eye boxes have long been major issues in holographic displays; however, considerable progress has been achieved in recent studies[2,4–7]. In contrast to the significant advancements in holographic displays, their counterpart, namely, the acquisition of holograms of the real world, has been less explored. Moreover, little

effort has been devoted to establishing a connection between hologram capture and display.

As holographic displays require hologram data as input, two main approaches are available for generating holograms of real-world scenes. The first approach involves capturing RGB-D images and calculating computer-generated holograms (CGHs)[8]; however, this method is heavily dependent on the accuracy of the depth map extraction process[9–11] or depth map measurements[12]. Improving the accuracy of a depth map typically requires extensive measurements and complex computations, which hinder the acquisition of high-quality depth maps in real time[13]. The second approach involves

[1]Samsung Advanced Institute of Technology, Samsung Electronics, 130 Samsung-ro, Suwon 16678 Gyeonggi-do, South Korea. [2]Department of Information Display, KyungHee University, 26, Kyungheedae-ro, Seoul 02447, South Korea. [3]Department of Electrical and Computer Engineering, Seoul National University, 1 Gwanak-ro, Seoul 08826, South Korea. [4]Present address: Department of Physics, Gachon University, 1342 Seongnam-daero, Seongnam, Gyeonggi-do 13120, South Korea. [5]These authors contributed equally: Hyeonseung Yu, Youngrok Kim. ✉e-mail: mins@khu.ac.kr; lhs12100@snu.ac.kr

directly capturing real-world holograms using holographic cameras. Holograms are typically captured using coherent laser light sources[14]; this approach has been particularly successful in biomedical imaging[15]. However, to capture real-world objects, the use of laser light is not practical as lasers present significant safety issues, especially when capturing human faces. Therefore, the development of an incoherent holographic camera[16,17] that captures real-world holograms using safe daylight is a promising path for the acquisition of real-world holograms.

Self-interference incoherent digital holography (SIDH) has been studied for decades following the incoherent hologram capture method proposed in ref. 18. The basic working principle of SIDH is to divide the light which is emitted or reflected from a single point into two waves using a wavefront division device and to modulate them differently to ensure that they can interfere at the image sensor plane. This concept is based on the fact that two split waves remain mutually coherent even under incoherent illumination because they originate from the same object point. SIDH has been implemented in various system configurations based on a polarization division approach[17,19] or the spatial division approach[16,20–22]. The polarization division-based approaches represented by Fresnel incoherent correlation holography (FINCH)[19,23,24] have been actively studied in biomedical imaging, and the recent developments have led to the commercialization of the system[25]. In contrast, extending the usage of SIDH systems to daily-use cameras, which is the main motivation of our work, has been relatively unexplored due to the difficulty of achieving a similar field of view (FoV) to general-purpose 2D cameras. We note that the SIDH systems optimized for imaging microscopic samples cannot be simply transformed into systems for imaging life-sized objects because the two systems have different optimal configurations: the former is designed to achieve high lateral resolutions[26] while the latter requires a moderately large FoV. Moreover, satisfying both requirements is challenging due to the trade-off between lateral resolution and FoV; a gap between the wavefront division device and the image sensor should be reduced to increase FoV, however, such a modification leads to the decreased lateral resolution[26].

Even if we consider only the system design choice of optimizing FoV at the expense of the lateral resolution, reducing the lateral resolution does not immediately lead to practical FoV in SIDH systems because the maximum FoV is limited by the minimum achievable gap between the wavefront division device and the image sensor. Conventional wavefront division devices such as liquid crystal on silicon (LCoS) spatial light modulators (SLMs)[27] or a combination of a spherical mirror and beam splitter[16] are implemented with the reflection geometry; therefore, the minimum possible gaps are still on the order of few centimeters due to the physical limitation of placing the optical components. Thus, some attempts to capture macroscopic 3D scenes[16,21] beyond the microscopic regime have been investigated; however, the FoV is limited to less than 3 degrees, mainly due to the large gap between the wavefront division device and image sensor.

Considering that a large FoV is essential for capturing life-sized objects, the recent development of SIDH systems based on geometric-phase (GP) lenses[28,29] appears to be the most promising direction for realizing general-purpose 3D cameras because the wide aperture of the GP lens and its compatibility with the transmission geometry enables an increased FoV. Furthermore, the negative and positive focal length pair induced by the GP lens supports reasonable lateral and axial resolutions (see Supplementary Information Section 2.3). However, optical imperfections in the GP lens introduce severe image degradation issues. Furthermore, correcting optical aberrations and imbalanced color weights at the system level is difficult because the GP lens is a passive component. Therefore, the computational approaches to overcome the image degradation issue should be developed to employ the GP lens-based SIDH systems for capturing daily 3D scenes.

In this work, we demonstrate a fully holographic streaming system, leveraging an incoherent holographic camera to acquire high-quality 3D holograms for holographic displays. Our work proposes a high-quality real-time holographic camera system that overcomes the poor image quality problem of incoherent holographic cameras that are designed for large FoVs by using a deep learning-based filtering technique. We consider GP-SIDH as the baseline system for the camera hardware and demonstrate that the employed neural network efficiently removes noise and enhances the image quality of incoherent holograms of various real-world scenes, including human faces. The proposed network is designed to operate with the complex-valued hologram data format throughout the processing pipeline, thus ensuring that the final outputs can be readily shown on holographic displays without any further CGH calculations. As the neural network handles single-shot holograms, multishot measurements for denoising via temporal averaging are not necessary. It should be noted that despite the development of single-shot capture systems[30–33], denoising has typically been performed via multishot measurements in SIDH systems[34–36]. Thus, by exploiting the real-time capture and processing capabilities and incorporating the high visual quality of the proposed deep learning-based incoherent holographic camera system, we realize a real-time holographic streaming system that acquires and displays real-world scenes on a holographic display based solely on hologram data. Our demonstration presents possibilities for developing practical holographic streaming systems or holographic teleconferencing systems.

## Results

### Deep learning-based incoherent holographic camera

As a key component in holographic streaming systems, we first demonstrate a deep learning-enabled, high-quality incoherent holographic camera system, i.e., DeepIHC. Our incoherent holographic camera system consists of GP-SIDH hardware[28] and a hologram filtering module, as shown in Fig. 1a. The GP-SIDH, in which the recording plane matches the sensor plane, is used to capture a raw hologram as shown in Fig. 1b. To reconstruct the object image, we propagate the raw hologram to the object plane using the angular spectrum method, as shown in Fig. 1c, and compute the intensity image, as shown in Fig. 1e. The poor image quality of the reconstructed image suggests that the raw hologram (Fig. 1b) captured with the GP-SIDH system alone cannot provide practically usable 3D data (see Supplementary Information Section 1 for more details on this system). The degraded image quality can be attributed to the hologram formation model. Captured incoherent holograms can be described as incoherent summations of impulse response functions from individual points in the captured 3D object. However, the impulse response functions in GP-SIDH systems have spatial and depth dependencies, which are highly challenging to characterize experimentally (see Supplementary Information Section 1.2). Moreover, even if we can acquire this information, 3D information about the target 3D object is required in the inverse correction of the optical aberrations, and this information is difficult to obtain. In addition to the image degradation introduced by the spatial variance in the impulse response functions, we observe that the signal-to-noise ratio (SNR) significantly decreases as the scene complexity increases (see Supplementary Information Section 1.2). To address the image degradation issue faced by existing incoherent holographic cameras, we propose a deep learning-based hologram filtering method as a postprocessing module of DeepIHC. Our main goal is to generate complex-valued holograms based on the captured holograms by using the neural network such that the focal images reconstructed from the generated holograms produce high-quality images while accurately reproducing the depth information. The proposed neural network outputs a noise-filtered hologram, as shown in Fig. 1d, and the image reconstructed from the filtered hologram shows a dramatically improved image quality, as shown in Fig. 1f.

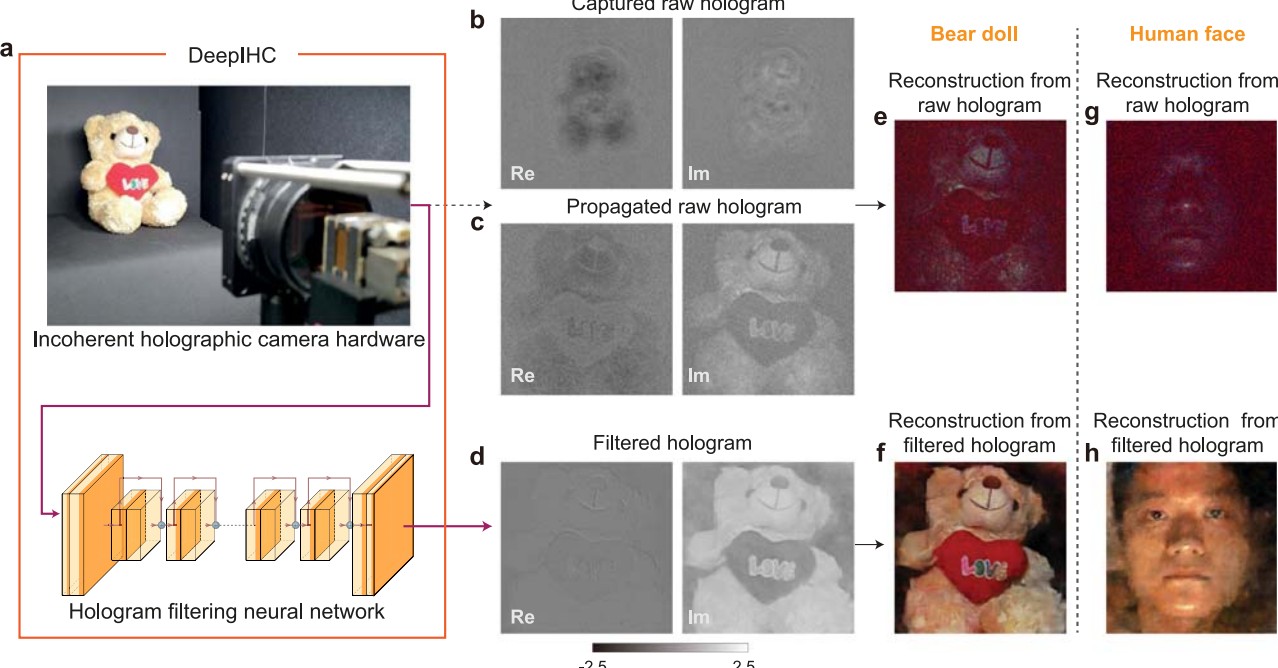

**Fig. 1 | Principle of the deep learning-based incoherent holographic camera (DeepIHC). a** Schematic of the deep learning-based incoherent holographic camera system. A hologram acquired by the geometric-phase self-interference incoherent digital holography (GP-SIDH) hardware is filtered by the proposed neural network. The filtering process operates in real time, and the output filtered holograms provide dramatic visual quality enhancements. **b** Raw hologram acquired by the GP-SIDH system. **c** Raw hologram propagated to the central plane of the object. **d** Filtered hologram inferred by the neural network. The real and imaginary parts are shown for only the green channel in **b-d**. Re: real, Im: imaginary. **e** Image of a bear doll reconstructed from the raw hologram in **c**. **f** Image of a bear doll reconstructed from the filtered hologram in **d**. The proposed system can capture human face holograms. **g** Image of a human face reconstructed from the raw hologram. **h** Image of a human face reconstructed from the filtered hologram.

Furthermore, while the identity of the human face in the focal image reconstructed from the raw hologram is almost unrecognizable in Fig. 1g, the face in the image obtained using the DeepIHC system is clear, as shown in Fig. 1h. Considering that the quality of the holograms acquired with the previous GP-SIDH system alone does not meet the practical requirements of 3D cameras, we can claim that the deep learning-based hologram filtering method enables the acquisition of practical 3D data that were not previously accessible.

Our proposed network architecture for hologram filtering is shown in Fig. 2a. The neural network is specifically designed to operate in a hologram-in hologram-out manner; the data formats of the input and output are both set to 6-channel 2D images, which consist of stacks of the real and imaginary parts of 3-channel color holograms. Whereas most denoising algorithms are applied to reconstructed 2D focal images[37–41] or intermediate light field representations[42], the proposed fully holographic processing pipeline provides two notable advantages: (1) the filtered output is a complex-valued hologram, which can be readily shown on holographic displays, and (2) pure holographic processing removes the need for intermediate representations such as RGB-D or light fields, thus reducing the computational complexity. To the best of our knowledge, our denoising algorithm is the first neural network proposed for denoising incoherent holograms that are acquired by incoherent holographic cameras.

We train the neural network via supervised learning, and we employ 2D images displayed on a 2D tablet as the reference 3D objects to acquire the dataset. Given that depth map acquisition is a challenging task that is actively being studied[11,43], our approach enables access to both the precise depth information of the given object and the ground-truth focal images required to compute the loss function, as the depth profile of the scene can be varied by simply placing the 2D tablet at different depth positions. It should be noted that employing holographic displays to generate reference 3D images is not a viable option because most holographic displays operate under coherent illumination conditions, which contradicts the working principle of incoherent holographic cameras. One major drawback of using fronto-parallel images is that the captured objects in the dataset contain only simple, flat depth profiles, which can lead to inaccurate results when handling occluded boundaries or multidepth scenes. However, the results indicate that this simplified approach can be extended to real-world scenes with complex depth profiles.

Figure 2b illustrates the proposed training procedure. When the target depth range is set to [30 cm, 48 cm] from the camera, a $1024 \times 1024$ hologram $H_{capture}$ of a target image displayed at a distance $d_i$ is captured and propagated to the central plane ($d_c = 39$ cm) using the depth-corrected angular spectrum method (d-ASM, see Methods). It should be noted that the hologram is propagated to the central plane regardless of the object depth. We chose this strategy because we cannot access the depth information of the captured objects during the validation stage.

$$H_{center} = f_{d-ASM}(H_{capture}, z = d_c). \quad (1)$$

This approach significantly reduces the receptive field size required by the neural network[8] and resolves the depth mismatches among the color channels. Then, a $720 \times 720$ subregion is cropped from the full hologram to obtain $H_{center}$. This cropping process considers two factors: the effective region of interest (ROI) of the system, which is limited to ~$600 \times 600$ (as shown in the later sections), and the sufficient margin of 120 pixels, which is set to prevent boundary artifacts that may occur during ASM propagation when diffracted beams diverge and contribute to nearby pixels. For the propagation within the [30 cm, 48 cm] range, the maximum expansion corresponds to 30 pixels in our system; however, we set the margin rather aggressively to

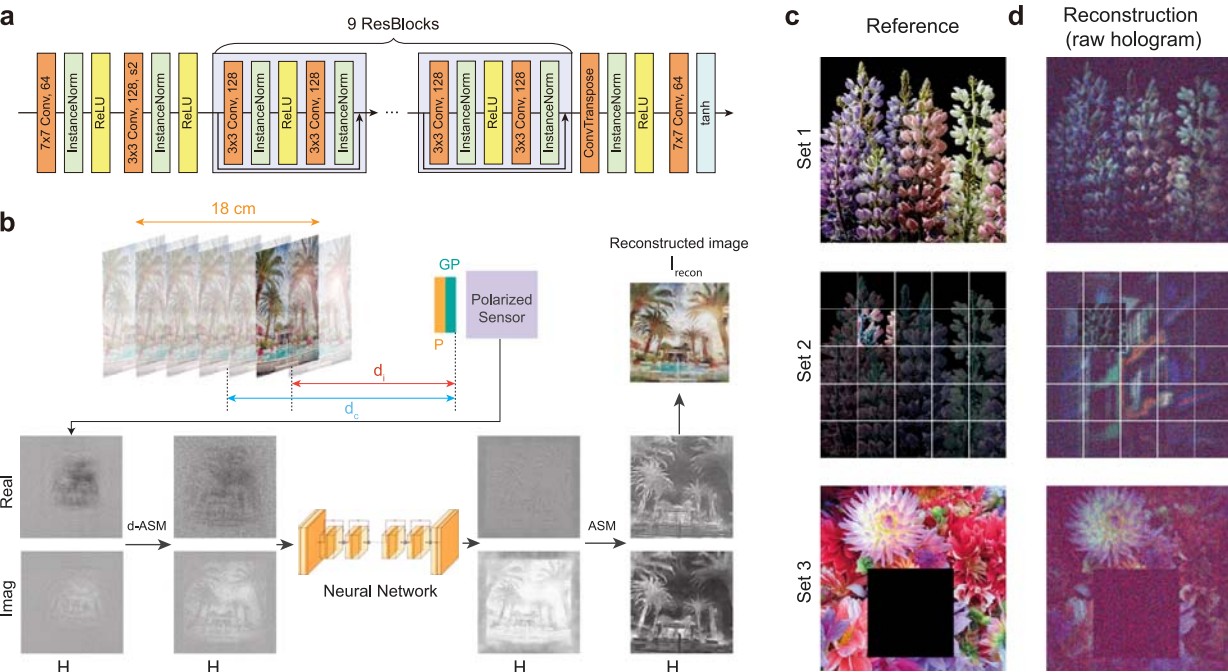

**Fig. 2 | Hologram filtering neural network. a** Neural network architecture for filtering holograms. The network contains one strided convolution block, nine residual blocks and one transposed convolution block. **b** Training procedure of the neural network. The real and imaginary parts of the holograms are presented only for the green channel. GP GP lens, P linear polarizer, $d_c$ distance between the central plane and GP, $d_i$ distance between the target image and GP, ASM angular spectrum method, d-ASM depth-corrected ASM. **c, d** Training dataset examples. **c** Reference target images displayed on a 2D display at various depth positions and **d** the corresponding captured holograms.

show that DeepIHC can handle a resolution of 720 × 720 in real time, as most high-quality media files support at least 720p resolution. The real and imaginary parts of the cropped $H_{center}$ are then stacked and fed into the neural network as input. The network outputs a hologram $H_{out}$ with the same format as the input hologram, and $H_{out}$ is propagated by an additional distance $d_o = d_i - d_c$ to generate $H_{recon}$, which represents the optical field at $d_i$.

$$H_{recon} = f_{ASM}(H_{out}, z = d_o) \qquad (2)$$

It should be noted that we propagate $H_{out}$ using the conventional ASM in this step, and all the color channels have the common propagation depth of $d_o$. Finally, we compute the perceptual loss[44] between the target image and the focal image $I_{recon} = |H_{recon}|^2$ reconstructed at depth $d_i$.

$$l_{pcp} = \frac{1}{W_{j,k}H_{j,k}} \sum_{x=1}^{W_{j,k}} \sum_{y=1}^{H_{j,k}} \left( \phi_{j,k}\left(I_i^{target}\right)_{x,y} - \phi_{j,k}(I_{recon})_{x,y} \right)^2. \qquad (3)$$

where $\phi_{j,k}$ is the feature map obtained by the $k$-th convolution layer before the $j$-th maxpooling layer in the VGG-19 network[44]. $W_{j,k}$ and $H_{j,k}$ denote the dimensions of the feature maps. We use the activation from the $VGG_{3,3}$ convolutional layer. Please refer to Supplementary Information Section 1.3 for the detailed procedure involved in matching the target image and $I_{recon}$. The neural network is trained for 120 h with 400 epochs using the Adam optimizer, and the batch size is set to 1.

To validate the trained neural network, we first test DeepIHC on a validation dataset consisting of planar images, as shown in Fig. 3. The images displayed on the tablet at various depths are presented in Fig. 3a, d, m and p, and their depths are indicated in the upper left corners of the images. Figure 3b, e, n, q presents the images reconstructed from the raw holograms at the corresponding depths using

d-ASM. Each color channel is separately renormalized to the range [0, 1] to balance the color channels. Compared with the target images, the reconstructed images from the raw holograms have poor image contrast and speckle noise, increasing the difficulty of perceiving fine details.

Figure 3c, f, o, r presents the images reconstructed from the filtered holograms acquired by DeepIHC. The proposed deep learning-based filtering method successfully restores the color appearance and drastically increases the image contrast in each image. The peak signal-to-noise ratio (PSNR) and structural similarity index measure (SSIM) also indicate that significant improvements are achieved over the method of reconstructing images according to the raw holograms. Although DeepIHC provides exceptional image enhancements, we observe that the quality of the image boundaries is inferior to that of the central region and that some details are removed. For example, the boat in the upper left corner of the ocean image in Fig. 3m is not present in the output of the proposed method in Fig. 3o. Since this detail is also missing the raw holograms in Fig. 3n, this tendency indicates that some information must be physically captured for the network to generate meaningful information. Moreover, the spatial resolutions of the images reconstructed using DeepIHC are slightly inferior to those of the ground-truth images. This result can be explained by the fact that the original holograms do not capture the target objects at high resolution due to the resolution limit of the GP-SIDH system (see Supplementary Information Section 2.1 and 2.2), which suggests that high spatial frequency signals must be physically captured for the neural network to restore fine details. In addition, the neural network should not rely heavily on image features for denoising because the captured holograms contain diffraction patterns rather than exact image features, and the main role of the neural network is to remove noise as opposed to generating images. Thus, we evaluate whether the neural network shows good denoising performance when

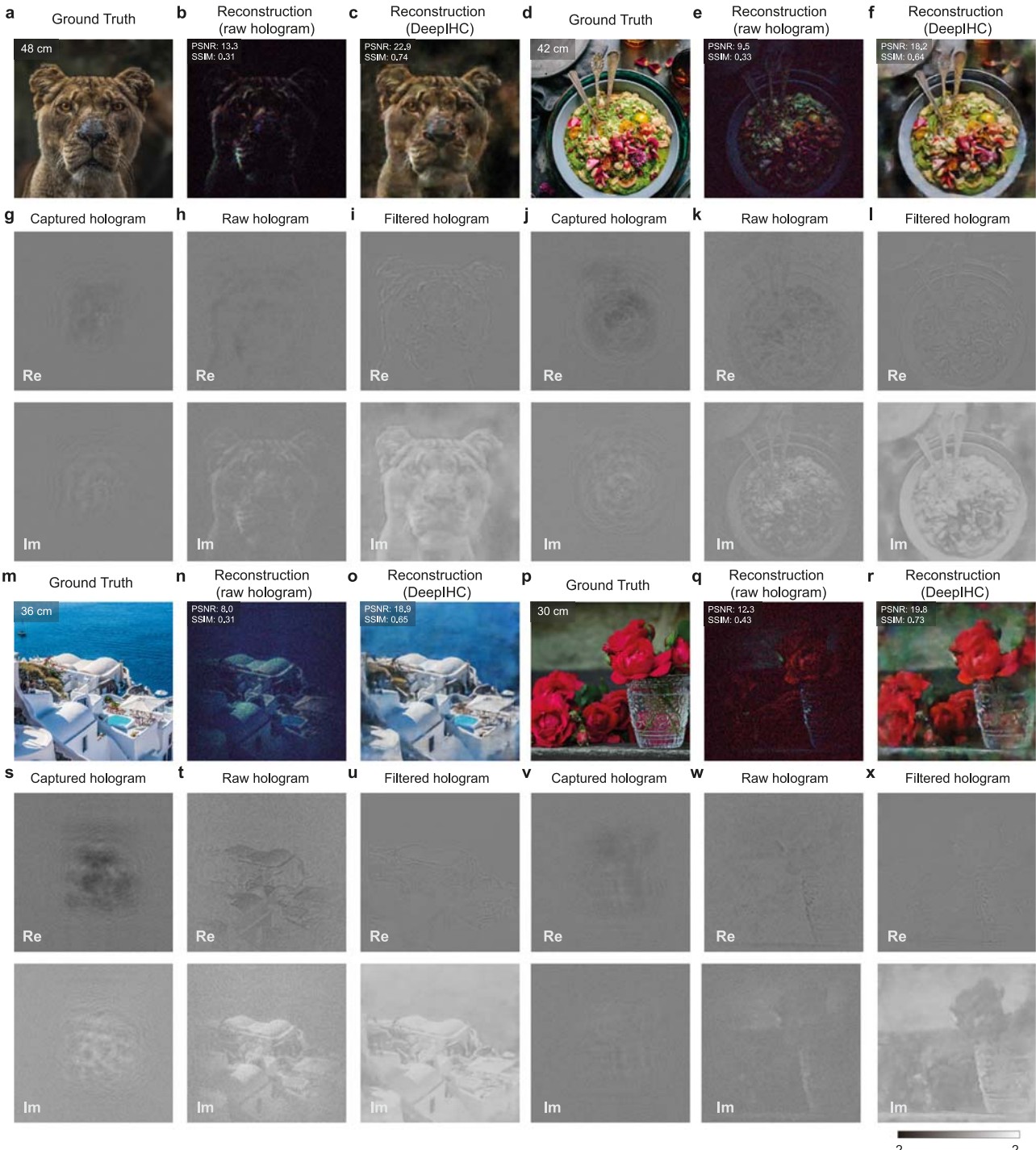

**Fig. 3 | Validation results of the hologram filtering neural network.**
**a**, **d**, **m**, **p** Target 2D validation images displayed at various depths. The object depths are specified in the upper left corner of each image. **g**, **j**, **s**, **v** Captured raw holograms. **h**, **k**, **t**, **w** Raw holograms propagated to the common depth $d_c$. **i**, **l**, **u**, **x** Filtered hologram obtained using the neural network. **b**, **e**, **n**, **q** Images reconstructed from the raw holograms at the target object depths. **c**, **f**, **o**, **r** Images reconstructed from the filtered holograms, which demonstrate exceptional quality advantages over the images directly reconstructed from the raw holograms. The resolution of each image is 600 × 600. The real and imaginary parts of the holograms are shown for the green channel only.

the validation image contains image features that did not appear during training, and we observe a similar improvement in the PSNR (see Supplementary Information Section 3.2).

In addition to the enhanced image quality, the essential and most important feature that DeepIHC should provide is the accurate reproduction of the depth information of the input hologram. To verify this capability, we test the focal stack computed from the holograms output by DeepIHC, as shown in Fig. 4d and h. For the

target image placed at $d = 48$ cm in Fig. 4a, the propagated raw hologram is shown in Fig. 4b and the filtered hologram obtained using DeepIHC is shown in Fig. 4c. Then, the images are reconstructed from the filtered hologram at three different depths, as shown in Fig. 4d. The image is best focused at the same depth as the target image, namely, at $d = 48$ cm (Fig. 4a), and the image gradually becomes blurred as the distance between the focus and image depth increases. Similarly, for the target image placed at $d = 30$ cm in Fig. 4e, the propagated raw

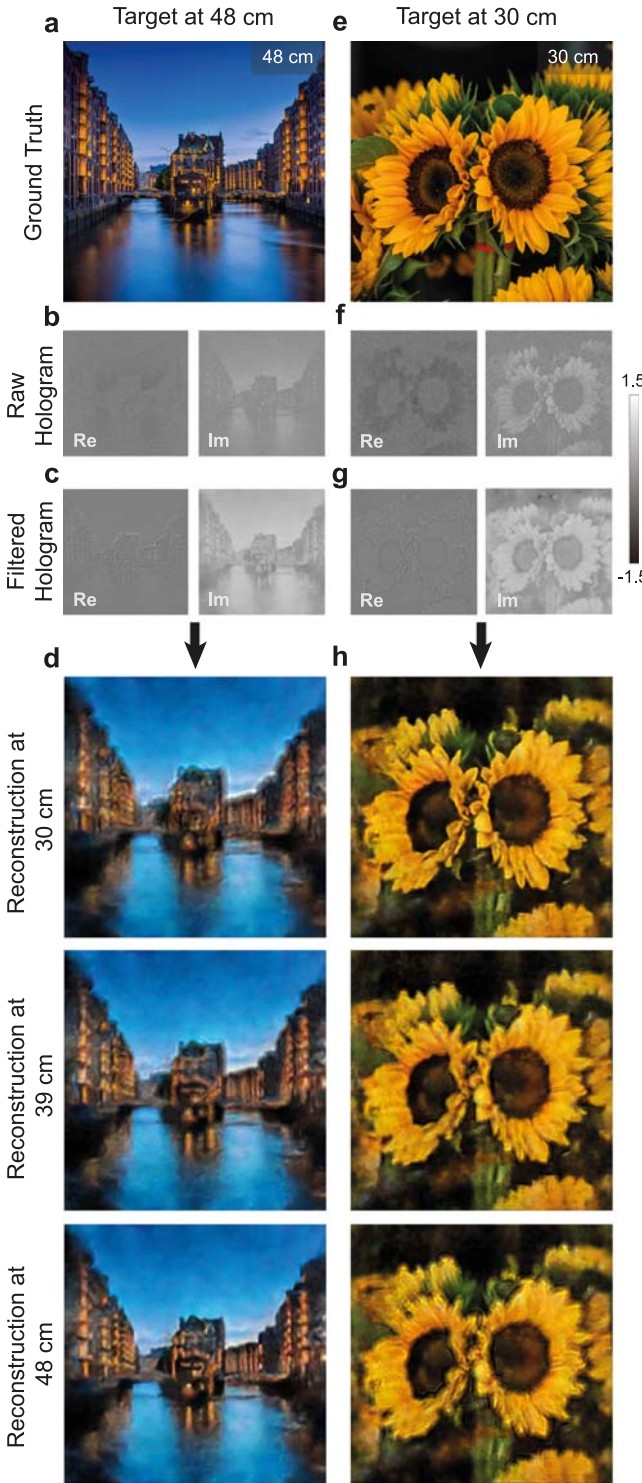

**Fig. 4 | Validation of the depth reproduction results of the hologram filtering neural network.** Target images at **a** 48 cm and **e** 30 cm were captured by DeepIHC to produce the raw holograms in **b**, **f** and filtered holograms in **c**, **g**, respectively. All holograms are shown for the green channel only. **d**, **h** Images reconstructed from the filtered holograms in **c**, **g** at different reconstruction depths. The night river view is best reconstructed at 48 cm, and the sunflower scene is best reconstructed at 30 cm, indicating that the object depths are accurately reproduced.

DeepIHC accurately reproduces the depth information of the target object. It should be noted that we do not provide any explicit depth information during the network inference stage. This indicates that the network preserves the phase information; the depth-dependent signals remain intact while the noise signals are effectively removed.

### Capturing real-world holograms

We capture several real objects with the DeepIHC system to test its generalizability beyond planar objects, and we find that our system produces visually enhanced holograms for nonplanar objects and multidepth scenes as well. Figure 1a shows our capture configuration; the real objects are placed within the [30 cm, 48 cm] depth range inside the FoV of the camera, and the objects are illuminated using a desk lamp. Figure 5 presents the testing results obtained for complex objects. For the mini statue scene (Fig. 5a) and miniature house scene (Fig. 5h), the captured raw holograms are propagated to the middle focus plane, and the focal images to which only simple normalization was applied are presented in Fig. 5e and l, respectively. The color reproduction in the mini statue scene is very poor, and only a few objects are observable in the miniature house scene. Figure. 5f and m present the reconstructed images derived from the DeepIHC holograms at the front, middle, and back focus. The front statue and background wall are separated by 15 cm in the mini statue scene, and the dog and back wall are separated by 8 cm in the miniature house scene. As the two scenes have different depth configurations, the depth values used in the focal image reconstruction process are indicated in the upper right corners of the images. For the mini statue scene, the neural network successfully handles the multidepth configuration without noticeable artifacts (Fig. 5f). Furthermore, the color information in the colored checker background is considerably better than that in the raw hologram. The enlarged views (Fig. 5g) exhibit clear defocus effects for the statues and the background. The neural network also successfully handles a scene with a more complex depth profile in a shorter depth range, as demonstrated by the miniature house scene (Fig. 5m). The enlarged views (Fig. 5n) show that the dog (front), ceiling light (middle) and round photo frame (back) are all accurately reproduced at their corresponding depths. It can be stated that DeepIHC reasonably accurately reproduces the color information in real-world scenes by considering the fact that even commercial 2D cameras produce different color appearances, and that the neural network is trained on only the color profile of the tablet screen.

### Real-time holographic streaming system and its applications

Based on the developed DeepIHC system, we demonstrate a real-time holographic streaming system that integrates DeepIHC and a holographic display prototype and operates with a refresh rate of 21 Hz. To the best of our knowledge, this is the first time that real-time acquisition and display of real-world holograms has been demonstrated. Figure 6a, b presents a schematic and photograph of the holographic streaming prototype, respectively. In our holographic streaming system, high-quality holograms acquired by DeepIHC are presented on the holographic display in real time. A validation camera with a variable focus is placed at one of the viewing positions of the holographic display to capture the displayed 3D scenes. Since the viewing area of the holographic display is limited to 5 mm, the displayed hologram is observed only by the validation camera in the viewing zone; therefore, it looks as if no image is displayed on the panel in the current photograph. Figure 6d, e presents the validation images of a merry-go-round music box scene that is captured by DeepIHC and shown on the holographic display. The front horse figure is focused in the front focal image, whereas the colored checker background is focused

hologram and filtered holograms are shown in Fig. 4f and g, respectively. Among the reconstructed images at three different depths shown in Fig. 4h, the best focus is observed at the same depth as the imaging target, namely, $d = 30$ cm (Fig. 4e). This result confirms that

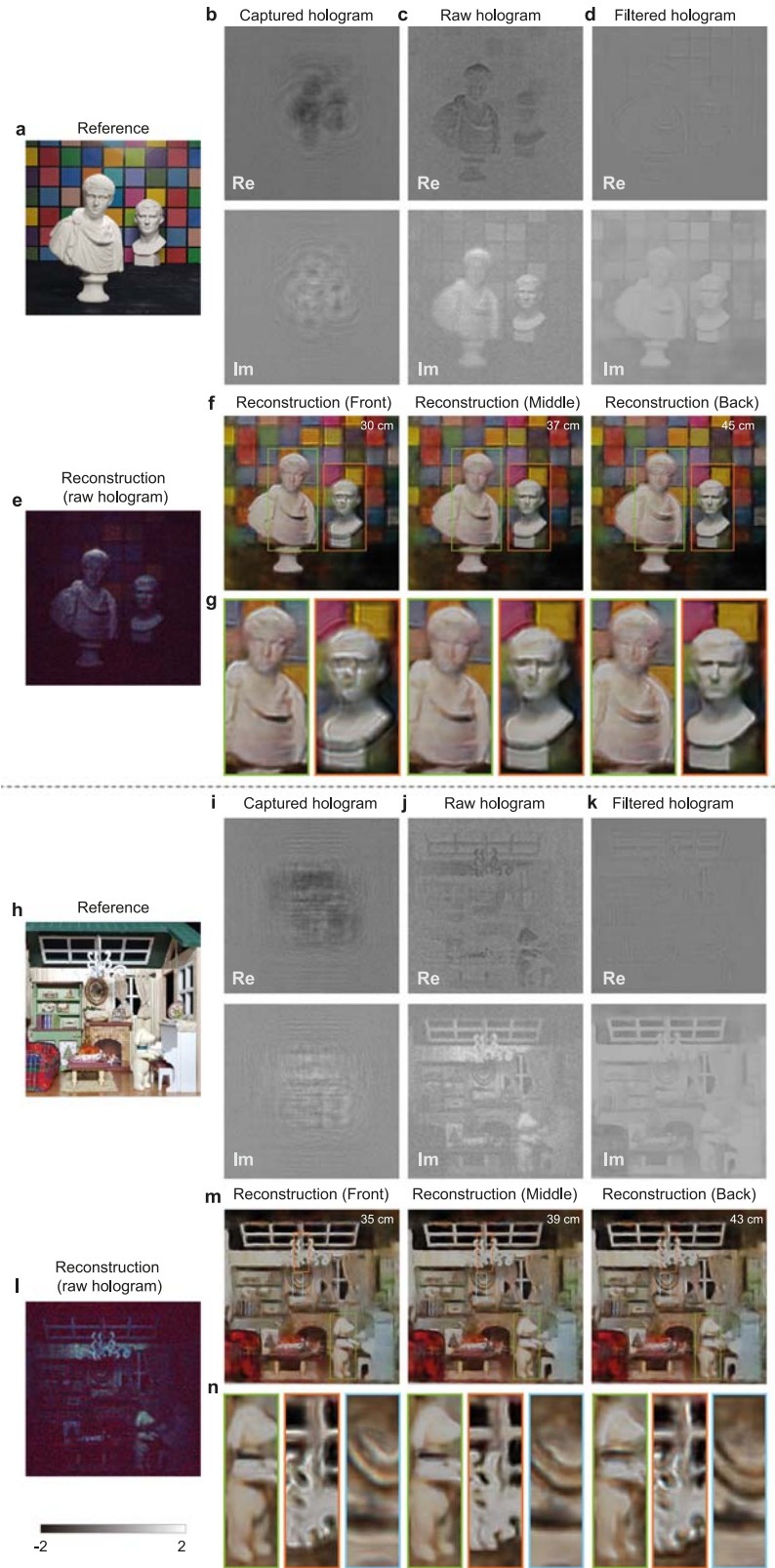

**Fig. 5 | Hologram filtering results for complex real objects. a, h** Reference photographs of the mini statue scene and miniature house scene, respectively. **b, i** Captured holograms and **c, j** propagated raw holograms at $d_c$. **d, k** Filtered holograms output by DeepIHC. **e, l** Images reconstructed from the raw holograms at the central object plane. **f, m** Front, middle, and back focal images reconstructed from the filtered holograms and **g, n** their corresponding enlarged views. All holograms are shown for the green channel only.

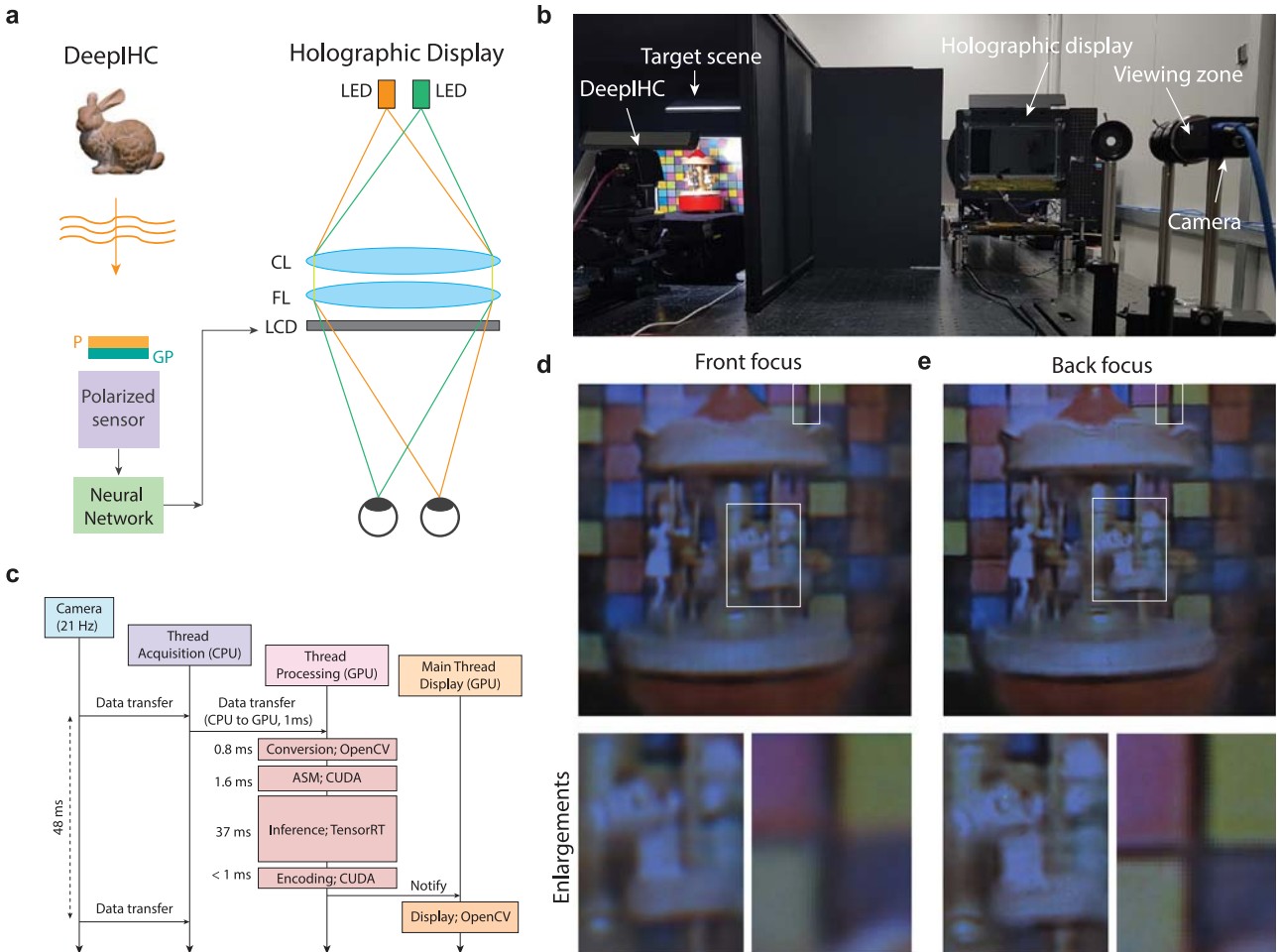

**Fig. 6 | Holographic streaming system. a** Schematic of the holographic streaming system. P polarizer, GP geometric phase lens, CL collimating lens, FL field lens. **b** Photograph of the holographic streaming system prototype. **c** Real-time hologram processing pipeline. The validation camera is placed in the viewing zone to capture the **d** front and **e** back focal images of the displayed hologram. Enlarged views of the horse figure and colored checker background are presented below the focal images.

in the back focal image. A reference photograph of the music box is shown in Supplementary Fig. S16. Supplementary Video 1 shows the real-time acquisition of the holographic images of the static music box with a variable focus. The frame rate of the validation camera was set to 2 Hz to ensure a sufficient exposure time due to the limited luminance of the holographic display. The accurate reproduction of the focal information on the holographic display is also observed in the video. Supplementary Video 2 shows the real-time acquisition of the holographic images for the moving music box. The camera frame rate was set to 30 Hz by increasing the gain level to demonstrate the real-time acquisition ability of the proposed system. Time-dependent noise signals are clearly observed in this case. The noise signals mainly originate from DeepIHC; however, noise is also induced by the high gain level. The lower noise level in Supplementary Video 1 than in Supplementary Video 2 suggests that the application of time-consistent denoising approaches[45,46] might reduce the flickering noise in the real-time streaming system.

After successfully demonstrating the proposed holographic streaming system, we explored possible applications of our proposed system. We note that teleconferencing is one of the most exciting applications of DeepIHC, as teleconferencing involves incoherent illumination conditions. Despite their practical importance, teleconferencing applications have not been extensively investigated in the context of holographic imaging due to

safety issues regarding the use of laser lights in coherent holographic imaging systems. As the DeepIHC system does not have these safety concerns, we demonstrate the real-time acquisition of human face holograms, as shown in Fig. 7. Supplementary Video 3 shows the video footage of this real-time acquisition. The details of the model's face are clearly resolved in the DeepIHC results, whereas the identity of the model is difficult to recognize based on the raw hologram results. In addition to the time flickering, which is similar to that observed in Supplementary Video 2, another aspect of DeepIHC is present: the image quality notably decreases as the face moves farther from the camera. This quality reduction occurs because the amount of light reflected from the face decreases as the distance between the face and the lighting increases. It should be noted that the target objects in our training dataset maintain the same brightness regardless of the object depth. Therefore, this trend indicates that the neural network must be trained on various lighting and capture conditions in the future. The raw holograms acquired in Supplementary Video 3 are also saved to a disk in parallel with the real-time streaming. By using the recorded hologram video, the videos of the focal images at three different focal depths are reconstructed for Supplementary Video 4. We observe that the in-focus plane changes as the face moves toward and away from the camera.

When the captured holograms are employed in AR applications, the introduction of editing can greatly expand their use,

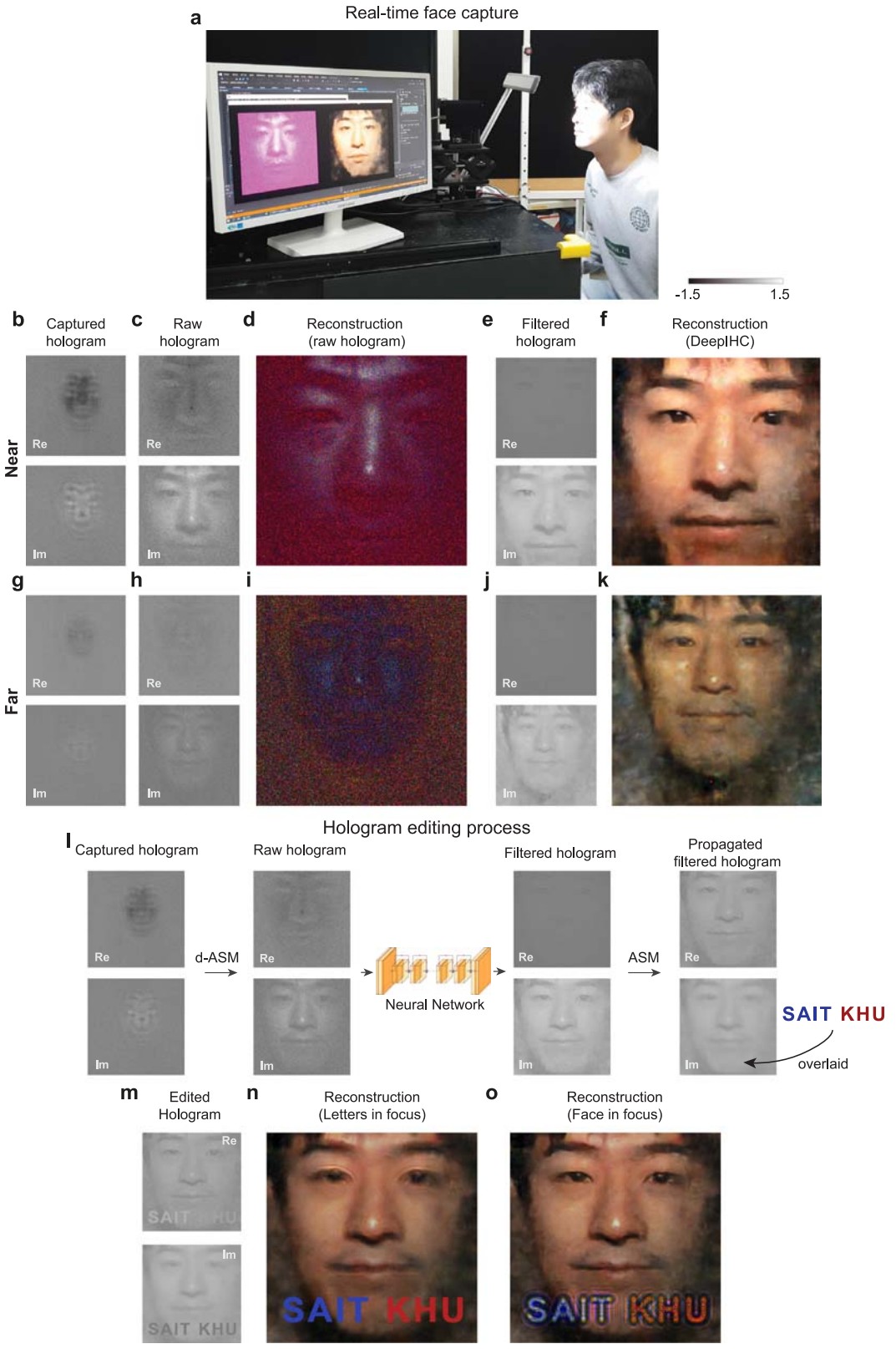

**Fig. 7 | Real-time face capture demonstration. a** Photograph of the real-time capture demonstration. **b**, **g** Captured holograms and **c**, **h** propagated raw holograms of the human face at the near and far positions, respectively. **d**, **i** Face images reconstructed from the raw holograms in **c**, **h**. **e**, **j** Filtered holograms obtained by using the neural network and **f**, **k** the corresponding reconstructed human face images at the near and far positions, respectively. **l** Hologram editing process. The filtered hologram output by DeepIHC is propagated to 25 cm in front of the face, and an artificial letter image is superimposed. **m** Resulting edited hologram. The images reconstructed from the edited hologram when the focus is set at **n** the letters and **o** the face. All holograms are shown for the green channel only.

such as in face augmentation, subtitle display and user interface presentation tasks. While the mixing of captured real-world holograms and artificial 3D objects requires in-depth investigations and is beyond the scope of this paper, we present a preliminary result of modifying the captured holograms with a simple text overlay. We propagate the hologram to 25 cm in front of the face and place the precomputed artificial letter images so that they occlude the face, as illustrated in Fig. 7l. This process does not violate the hologram formation model, as the contributions from point sources are incoherently summed. The images reconstructed from the edited hologram (Fig. 7m) at the text plane (Fig. 7n) and face plane (Fig. 7o) show that the captured hologram and artificial images are seamlessly blended.

## Discussion

In this work, we demonstrate a holographic streaming system as a step toward developing the ultimate holographic ecosystem in the future. As the key component in the streaming system, we propose a deep learning-based incoherent holographic camera system that filters noise and enhances the visual quality of incoherent holograms at a refresh rate of 21 Hz. We validate the enhanced visual quality of the images produced by this system for various 3D scenes, including planar photographic images placed at various depths and scenes with multi-depth objects. Since the proposed system is designed to output complex holograms, the filtered holograms can be shown on holographic displays with a simple encoding step. Moreover, we demonstrate the capture-to-display pipeline in real time, and the use of incoherent illumination allows for the acquisition of human face holograms.

Several interesting issues should be considered to improve the performance of the holographic streaming system. Although we drastically improve the image quality of the incoherent holographic camera, the hardware system should be enhanced so that it can be widely applied as a practical 3D camera. The low spatial resolution of the system due to the limited aperture and sensor pixel density reduces the resolution of the fine details in the acquired 3D scenes. To address these issues, we can employ multiple cameras to increase the effective aperture[34] because the sensor area defines the aperture size in the GP-SIDH system. Complementary metal-oxide semiconductor (CMOS) cameras with higher pixel densities are also highly desirable, as they prevent aliasing effects and can capture the high spatial frequency components of incoherent holograms. To expand the current depth range of [30 cm, 48 cm], a training dataset at an extended depth range should be collected, and the low light collection efficiency of the GP-SIDH system should be increased. Extending the depth range requires a neural network with a larger receptive field; therefore, large propagation kernels[47] beyond those standard convolution layers used in DeepIHC should be investigated. We also found that the diverse depth configurations in real-world scenes are challenging to incorporate during neural network training due to the difficulty of extracting precise depth information in arbitrary 3D scenes. Therefore, a new strategy for collecting fully 3D real-world datasets with appropriate RGB-D reference data should be devised.

In relation to holographic displays, several considerations should be examined when implementing practical holographic streaming systems. To support wide viewing angles or eye boxes in holographic displays, spatial light modulators with high pixel densities must be developed. These devices would require more dense information about incoherent holograms, therefore the required amount of information in practical settings and the handling of such data in streaming systems should be investigated. This issue also motivates the further optimization of the computational time of the neural network, as the system is still slow considering that the neural network produces $720 \times 720$ holograms. Extending the proposed network to higher resolution holograms is straightforward, as it is a fully convolutional network. However, the inference time typically increases with the input image size. Therefore, an optimal neural network architecture must be developed to support the generation of full high-definition (FHD) or ultra HD (UHD) holograms in real time. Although our work is inspired by the recent development of learned hologram generation methods[5,6], we did not consider optimizing the filtered holograms for specific holographic displays and instead focused on the different goal; the acquisition of high-quality holograms of real-world scenes. In future works, it would be interesting to explore how to optimize the holograms output by DeepIHC for actual holographic displays. This research direction poses a new challenge because the basic assumption of the learned hologram generation method, namely, that the depth information is already known, does not hold for incoherent holographic cameras, as the depth information is implicitly encoded in incoherent holograms.

Despite these challenges, we believe that our work demonstrates an important milestone in holography research: the realization of a holographic streaming system, showing that the existing 2D video streaming systems can be realized in a fully 3D holographic manner. Our work paves the way toward the ultimate holographic ecosystem and would inspire the development of holographic broadcasting systems or holographic teleconferencing systems in the future.

## Methods
### GP-SIDH
The system configuration is shown in Supplementary Fig. S1. Our system employs a custom-made GP lens (ImagineOptix) with focal lengths of $f_p = 1000$ mm and $f_n = -1000$ mm, and $d_s$ is set to 6 mm. All holograms are captured with single-shot measurements, while the original GP-SIDH system in the reference[28] averages multiple images to increase the SNR.

### Image reconstruction
The captured hologram $\mathcal{H}$ can be optically propagated similarly to a conventional CGH, and the ASM[48] is selected as the propagation algorithm in our study. One notable difference between the holograms captured by the GP-SIDH system and conventional CGHs is that each color channel has a different propagation depth due to the chromatic characteristic of the GP lens[49–51]. Therefore, we perform the d-ASM to reconstruct a focal image at depth z as follows:

$$f_{d-ASM}(\mathcal{H}, z; \lambda) = \iint \mathcal{F}(a(x, y, \lambda) e^{i\phi(x,y,\lambda)} \mathcal{H}(x, y))$$
$$\cdot \mathcal{K}(f_x, f_y, \lambda, z_\lambda(z)) e^{i2\pi(f_x x + f_y y)} df_x df_y \tag{4}$$

where

$$\mathcal{K}(f_x, f_y, \lambda, z_\lambda(z)) = \begin{cases} e^{i\frac{2\pi}{\lambda}\sqrt{1-(\lambda f_x)^2-(\lambda f_y)^2}z_\lambda}, & \text{if } \sqrt{f_x^2 + f_y^2} < \frac{1}{\lambda}, \\ 0 & \text{otherwise} \end{cases}$$

and

$$z_{\lambda_r} = z\frac{\lambda_g}{\lambda_r}, \quad z_{\lambda_g} = z, \quad z_{\lambda_b} = z\frac{\lambda_g}{\lambda_b}.$$

Here $f_{d-ASM}$ denotes the depth-corrected propagation operator; $f_x$ and $f_y$ represent the spatial frequencies; $\mathcal{F}$ denotes the Fourier transform operator; $\mathcal{K}$ denotes the transfer function. $a(x, y)$ represents a constant function; and $\lambda_r$, $\lambda_g$ and $\lambda_b$ denote the red, green and blue wavelengths, respectively. The propagation distances are calibrated with respect to the green wavelength. For the holograms in which the depth mismatches between different color channels are already compensated, the same propagation lengths should be used for each color channel. In this case, we perform the conventional ASM, denoted by $f_{ASM}(H, z)$, where $z_{\lambda_r} = z_{\lambda_g} = z_{\lambda_b} = z$.

## Dataset

Some example images in the training dataset are shown in Fig. 2c. In our dataset capture process, we consider seven equally spaced depth planes spanning 18 cm corresponding to $d \in [30 \text{ cm}, 48 \text{ cm}]$, where $d$ denotes the distance between the object and the GP lens. Three sets of 250 holograms were acquired at each depth $d_k$. Set 1 was collected by displaying 250 images $I_i, i \in [1, 250]$, from the DIV2K dataset[52] by applying only a simple cropping process and capturing the corresponding holograms: $H_i^{(1)}(d = d_k)$. Set 2 is the augmented dataset, which simulates a multidepth scene dataset that was generated without capturing additional images. For each hologram $H_i^{(1)}(d = d_k)$ in Set 1, a subpatch is randomly selected from the $5 \times 5$ grid; the remaining region is then replaced by a randomly selected $H_j^{(1)}(d = d_j)$, where $i \neq j$ and $d_k \neq d_j$ with a margin of 20 pixels are used to obtain $H_i^{(2)}(d = d_k)$. The loss function is computed for only the selected subpatch region in this case. The holograms in Set 3 are captured for images, $I_i, i \in [251, 500]$, in the DIV2K dataset. These images contain null regions that help the network to efficiently learn dark backgrounds. The height and width are independently and randomly selected between 0.25 and 0.5, assuming an image size of 1, to maintain reasonably sized dark regions. Various depths share the same set of target training images to ensure that the network effectively learn the differences in images acquired at various depths. The 2D images are displayed using a 12.9-inch tablet screen. The total time required to capture the training dataset was 12 h. The authors affirm that human research participants provided informed consent for publication of the images in Figs. 1 and 7.

## Holographic display system

Figure 6a presents a schematic diagram of the proposed holographic streaming system. The holographic display system was built based on a flat-panel display type[53]. The lights from two LEDs (Doric Lenses Inc. w55) was collimated by a custom lens ($f = 50$ cm) and focused by using a custom field lens ($f = 1$ m). A commercial 10.1-inch LCD panel (BOE, TV101QUM-N00-1850) with a resolution of $3840 \times 2160$ and a pixel pitch of 58.05 μm was used to encode the complex hologram through amplitude-only modulation[54]. For the given complex hologram $H_{proj}$, the corresponding pattern $P_{proj}$ to be shown on the holographic display was calculated as follows:

$$P_{proj} = \mathbf{Re}(H_{proj}) + |H_{proj}|. \tag{5}$$

The viewing distance was set to 1 m, which is equal to the focal length of the field lens. The two waves that originate from the two LEDs were projected onto the left and right eyes, and the interpupillary distance was adjusted by changing the separation between the two LEDs. Although our display supports stereoscopic views, we projected the hologram onto only a single viewpoint in our demonstration.

## Real-time processing

Figure 6c depicts the real-time hologram processing pipeline. The camera (Lucid Vision Labs PHX050S-QNL) operates at 21 Hz, and each frame is continuously acquired via the acquisition thread. The raw data are uploaded to the GPU at this stage to reduce the processing time. The hologram data are then retrieved using the OpenCV CUDA module, which requires the deinterleaving computation shown in Eq. 9 in the Supplementary Information and bilinear demosaic processing. The d-ASM operation is then performed using the cuFFT library to compute the hologram at the central plane. The ASM computation process requires Fourier transformations, multiplication with the precomputed ASM kernels, and inverse Fourier transformations. The next step involves the computation of the filtered hologram using the neural network. The network inference process uses the TensorRT module with fp16 precision, and the execution time is 37 ms. Finally, the output hologram is encoded as a suitable display format. The amplitude encoding of the complex hologram is computed in the proposed holographic streaming demonstration based on Eq. (5), and the focal image at the central plane is computed for the face video capture demonstration. Both encoding methods present negligible execution times of less than 1 ms. The final output images are displayed using the OpenCV module with OpenGL support in the main thread. The overall data transfer and processing time is ~40 ms, which is less than the acquisition time interval of the camera (48 ms). Therefore, the proposed system operates at 21 Hz and is limited only by the frame rate of the camera. Although the system latency is not exactly calibrated, a latency of less than 100 ms is expected according to the time profile. The system is implemented in C++ based on the interoperability between TensorRT, CUDA, and OpenCV on a GPU, and all execution times are measured on an NVIDIA RTX3080.

## Data availability

All relevant data that support the findings of this work are available from the corresponding author upon reasonable request.

## Code availability

All relevant codes that support the findings of this work are available from the corresponding author upon reasonable request. See Supplementary Information Section 4 for pseudocodes.

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

## Acknowledgements

This work was supported by Samsung Research Funding & Incubation Center of Samsung Electronics under Project Number SRFC-IT2201-03.

## Author contributions

H.Y. designed and implemented the filtering algorithm and streaming system, performed the experiments and wrote the manuscript. Y.K. developed the camera system, performed the experiments, and wrote the manuscript. Y.K., D.Y., and W.S. were involved in developing the proposed algorithm. The experiments were performed with help from Y.K., J.-Y.H., H.S., and G.S. Y.S. reviewed the manuscript. S.-W.M. and H-S.L. supervised the overall work.

## Competing interests

The authors declare no competing interests.
