## [Peer Review File · Nature Communications]

Deep learning-based incoherent holographic camera enabling acquisition of real-world holograms for holographic streaming systemREVIEWER COMMENTS

Reviewer #1 (Remarks to the Author):

Review of "Towards holographic streaming systems: From real-time acquisition of the real world to holographic display."

The authors have proposed an interesting method for real-time acquisition and display of incoherent scenes using SIDH implemented using GP lens. There are many gaps in the description and many important references missing. The results could be improved to the level of images shown in celloptic. I believe the problem is the GP lens.

Major comments

1. The concept of SIDH is not explained well. It is written "The basic principle of SIDH is to split the incident light into two waves that are modulated differently so that they can interfere at the image sensor plane." SIDH works on a completely different holographic principle unlike coherent holography. It is not possible to create hologram by interfering light diffracted from two different points. In SIDH, the hologram is formed by self-interference i.e light emitted from a point is split into two and interfered. This must be stated in the first place of description and the difference with respect to coherent holograms must be clearly stated.
2. The reference 15 is not the original one. The original references are the conoscopic holography by Sirat and Rotational shear interferometer. With active device the first report was published by Rosen and Brooker in 2007 as FINCH - Joseph Rosen and Gary Brooker, "Digital spatially incoherent Fresnel holography," *Opt. Lett.* 32, 912-914 (2007) and there were many reports on FINCH. The work of Kim was published much later in 2013 with Michelson interferometer configuration.
3. It is not clear what it is meant "however, the capabilities of these approaches are limited due to low depth resolutions". Do you mean low axial resolution? Axial resolution can be controlled in FINCH and SIDH by changing the beam overlap conditions. Unless the maximum lateral resolution is needed for this application where the two beams are perfectly overlapped, the overlap condition can be varied to improve axial resolution.
4. Once again reference 19 for geometric phase lens is not original. There are numerous articles on Geometric phase lens. What about the *Nat. Photon* article on CINCH where the birefringent lens was used to demonstrate FINCH? The seminal work in *Nat. Photon* 2008 on FINCH was also not referred. It can be seen that the random multiplexing as shown in the above mentioned article, still works for sparse objects. BTW the geometric phase lens is not a good direction to proceed especially due to the low beam overlap condition as clearly explained in the many articles on FINCH and SIDH. It has advantages but many disadvantages also. The shortened optical path can be achieved using dual lens FINCH with no loss. The GP approach achieves this but with loss – reduced overlap. I suggest the authors to do a thorough literature review and add relevant important articles. Many relevant articles by Nomura, Nobukawa, Tahara, Matoba and Juodkazis have not been discussed. After that description, the authors followed "However, poor image quality has remained a key unresolved issue in these studies" Can you

explain the origin of this issue? Poor quality? Why? How? Unresolved? How? FINCH and SIDH are well-established techniques and there is no unresolved issue. For every problem, a solution has been found and published as articles. The only remaining problem in FINCH and SIDH is the need for multiple camera shots. Solutions are available with a choice of imaging characteristics to sacrifice – field of view, snr etc. The high FoV of GP needs to be explained.

5. In the manuscript, it is repeatedly mentioned – poor image quality which is not true. Yes in the 2013 article. But not in 2022. The current results of FINCH and SIDH are significantly better now and even commercialized by celloptic. Imaging results are available in their site.

6. Again it is mentioned “As the neural network handles single-shot holograms, we also eliminate the need for multishot measurement, which has been typically practiced to reduce noise, hindering real-time applications” multishot measurement has been already eliminated by many methods. This is not eliminated by the authors by Deep learning for the first time. Even without deep learning, it is possible to achieve single shot in FINCH and SIDH. Please do a thorough literature survey and add relevant articles on FINCH and SIDH.

7. The raw hologram pictures shown in Fig. 1b do not appear as FINCH/SIDH holograms. Can you show the recording plane in figure? Why for raw hologram and filtered hologram two images are shown? Why both have the image of the object 1 and not the object 2. Was a color camera used for recording? In FINCH and SIDH during reconstruction, the colour information is reconstructed at different locations due to the chromatic aberration introduced by diffractive components. How this problem was handled in this case? color based propagation still results in some error.

8. Please provide the images of hologram for all cases. The visibility of the holograms decrease with increase in spectral width. This is an important challenge. How this was rectified in your approach?

9. Please explain the resolution limit - lateral and axial as per the GP overlap conditions.

10. The image of the letters are ok but the image is noisy. Please provide holograms for all cases.

11. The limits on the AI method needs to be addressed.

Minor

1. There is a typo in incoherent in Fig. 1

Reviewer #2 (Remarks to the Author):

The submitted manuscript, "Towards holographic streaming systems: from real-time acquisition of the real world to holographic display", introduces a learned algorithmic approach for a holographic camera system. Specifically, the holographic camera operates with incoherent light, and the learned algorithmic approach helps retain visual quality by denoising.

A conclusion of my review could be found at the end of this text.

Typos, Style, Grammar

- Please consult with a native speaker. In the title, you may need to be saying: "from the real-time acquisition of real-world".

- Line 91-94, your line 87 already identifies the poor image quality problem.

- I treat figures as mini-pages. Please check with the common culture in the journal you are submitting for. I trust that you may need to write acronyms in full at first appearance in a figure's caption. Otherwise, readers must visit various places in the manuscript to understand each term. It degrades readability, in my opinion.

Comments

- Line 50, none of the works in 2 or 4-6 deals with narrow eye box issues, but they aim for visual quality. Consider citing: Kuo, Grace, Laura Waller, Ren Ng, and Andrew Maimone. "High resolution étendue expansion for holographic displays." *ACM Transactions on Graphics (TOG)* 39, no. 4 (2020): 66-1.

- Line 104, you may consider highlighting multishot as helping with temporal averaging of noise. I trust that temporal averaging is more common in computer-generated holography communities than multishot learning.

- In figure 1 and your introduction, I gather that your previous work is geometric-phase lens-based self-interference incoherent digital holography. Your contribution is to derive a learned hologram capture algorithm.

- Line 122-124, the statement is not clear and came out of the blue. What is the problem with the spatial variance of the impulse response function?

- Line 125, if I gather correctly, as the number of points to resolve in a scene increases, the captured data gets highly convoluted. Thus, you can no longer accurately pinpoint a reliable solution with a forward and inverse model. Sounds fair.

- Line 129-131, would it be fair to call your learned method a denoiser? It looks like you already agree in lines 146-147.

- In line 166, in my view, you need to dedicate a paragraph on how your camera hardware works actually. Because your readers are left with a learning routine and capture that they will have a hard time following.

- Line 166-170, this part needs a little bit of decyphering. You capture a hologram that represents a volume bounded by propagation distances ranging between 30 to 48 cms. Your hologram captures are 1024 by 1024 pixels. Your hologram represents the propagation of d_i , but in code, you propagate it to the central plane d_c using the Angular Spectrum Method.

- Line 170. Is there a color-independent angular spectrum method? I always thought light propagation kernel in any approach is wavelength dependent, no?

- Line 171-172. Yes, when you propagate to large distances, the kernel size of your light propagation simulation would be larger. Yes, this would translate to a need for larger receptive fields in a learned algorithm. If I follow correctly, you suggest that having the hologram at the center of the volume minimizes the requirement in large kernel sizes. Let me highlight that there are also large kernel-based learned algorithms that do not follow classical five-by-five or 11 by 11 convolutional layers: Kavakli, Koray, Hakan Urey, and Kaan Akşit. "Learned holographic light transport." Applied Optics 61.5 (2022): B50-B55.

- Line 177, are these boundary artifacts related to aliasing?

- Line 186, which perceptual loss are you using, though? [29] explains several, no?

- Line 186, I strongly suggest the authors release a public code repository as their work gets accepted.

- Line 186-187, where do we get the ground truth images for the images captured in holograms? Is there a secondary camera in your system that captures conventional photographs of a scene, and if so, how do you deal with the transformations required between a holographic camera and a conventional camera. Later, I read section 3 of your supplementary, and I gather that you are using a display as a proxy and mapping what is on the screen to ground the truth image. This is fair, but help me, please also understand, wouldn't that mean you are dealing with a two-dimensional hologram dataset but not three-dimensional captures? How does your network generalize to the case of three-dimensionality in this case? Or is this system fixed focus? Do you still need to adjust the focus manually, as in the supplementary video?

- Line 190, how many holograms do you have in total in your train set? Will you be able to release this set in the future publicly? Do you have large epochs because you have a small dataset?

- In lines 211-212, convolutional layers tend to "hallucinate" things. When you test your network against a test validation, were there cases where you have an image with a feature that has non-existing in the training dataset? Say if, in your train dataset, there is no a boat or a chair, etc., would it still denoise properly? I encourage the authors to add such a case to their comparison in figure 3.

- Across the figures, besides Figure 2, you don't show a hologram you captured, right? So when you say "raw hologram", you mean "classical reconstruction from a raw hologram", am I correct?

- Line 269, what does this holographic display entail?

I am unaware of a similar complete holographic display system in the literature. Thus, I believe the provided system is novel in that respect. On the other hand, if I gather correctly, the primary contribution here is the learned algorithmic approach, a series of convolutional layers followed by activation blocks. Furthermore, this specific network operates only in the hologram domain; both input and output are holograms. Previous works cited in their literature consist of networks that only work on the hologram domain (hologram in, hologram out; here [5] U-Net takes a hologram and spits out a hologram). I value this research and understand the importance of the research work. However, I need to be convinced. I trust the authors can help me by carefully addressing my comments in their responses. As is, I am leaning towards rejection.

Reviewer #1 (Remarks to the Author):

Review of "Towards holographic streaming systems: From real-time acquisition of the real world to holographic display."

The authors have proposed an interesting method for real-time acquisition and display of incoherent scenes using SIDH implemented using GP lens. There are many gaps in the description and many important references missing. The results could be improved to the level of images shown in celloptic. I believe the problem is the GP lens.

We thank the reviewer for acknowledging that we proposed an interesting system based on GP-SIDH. We regret that some of our descriptions and references were not complete and precise, but we believe they do not impact the novelty and contribution of our work. We put our best efforts into addressing the reviewer's concerns and adequately positioning our work among the previous works in the revised manuscript by reflecting the reviewers' suggestions.

Regarding the image quality of GP-SIDH against Celloptics, we believe this gap comes from the fact that those systems are optimized for different purposes: imaging macroscopic scenes and microscopic objects. For imaging daily scenes, we need to provide a moderately large FoV to capture multiple objects or large-scale objects. In contrast, for imaging biological cells, enhancing the resolution is one of the most important goals, and developing super-resolution techniques have been the core research field over the decades. We believe the reviewer would agree with us that achieving both large FoV and high resolution simultaneously is an extremely challenging task in FINCH/SIDH. Therefore, we would like to highlight that we aim to solve the image quality issue for the particular system setting which is optimized with more emphasis on large FoV.

In designing a SIDH system with larger FoV, the GP lens provides several benefits (please refer to our answer to Q4), but as suspected by the reviewer, the GP lens has an intrinsic limitation regarding the image quality: due to its passiveness, we cannot apply the techniques to actively compensate the defect or aberrations of the passive optical element unlike in FINCH. One of our motivations is to fully benefit from the use of the GP lens in the context of using the SIDH as a general-purpose camera, therefore we developed the data-driven computational approach to overcome the physical limitation of the passive GP lens.

Major comments

1. The concept of SIDH is not explained well. It is written "The basic principle of SIDH is to split the incident light into two waves that are modulated differently so that they can interfere at the image sensor plane." SIDH works on a completely different holographic principle unlike coherent holography. It is not possible to create hologram by interfering light diffracted from two different

points. In SIDH, the hologram is formed by self-interference i.e. light emitted from a point is split into two and interfered. This must be stated in the first place of description and the difference with respect to coherent holograms must be clearly stated.

We thank the reviewer for bringing our attention to this point. As suggested by the reviewer, we find that the description 'split the incident light into two waves' can be misleading, so we modified the sentence as follows in the revised manuscript.

Line 72-75: *"The basic working principle of SIDH is to divide the light which is emitted or reflected from a single point into two waves using a wavefront division device and to modulate them differently to ensure that they can interfere at the image sensor plane."*

2. The reference 15 is not the original one. The original references are the conoscopic holography by Sirat and Rotational shear interferometer. With active device the first report was published by Rosen and Brooker in 2007 as FINCH - Joseph Rosen and Gary Brooker, "Digital spatially incoherent Fresnel holography," Opt. Lett. 32, 912-914 (2007) and there were many reports on FINCH. The work of Kim was published much later in 2013 with Michelson interferometer configuration.

As our system mainly aims for the usage of SIDH as a general-purpose camera, we cited reference 15 as a representative paper demonstrating that SIDH can capture daily scenes, whereas FINCH and other works mainly focus on capturing microscopic objects. As we are also aware of the significance and importance of the suggested work, we added the reference to the revised manuscript.

3. It is not clear what it is meant "however, the capabilities of these approaches are limited due to low depth resolutions". Do you mean low axial resolution? Axial resolution can be controlled in FINCH and SIDH by changing the beam overlap conditions. Unless the maximum lateral resolution is needed for this application where the two beams are perfectly overlapped, the overlap condition can be varied to improve axial resolution.

As pointed out by the reviewer, we can control the axial resolution by changing the overlap condition in both FINCH and SIDH. As FINCH or CINCH have evolved towards achieving higher resolution or super-resolution to capture fine details of microscopic samples, there have been additional attempts to increase the axial resolution such as increasing axial resolution via a spinning disk confocal system [N. Siegel and G. Brooker, "Improved axial resolution of FINCH fluorescence microscopy when combined with spinning disk confocal microscopy," Opt. Express 22, 22298–22307 (2014)]. As discussed, the lateral and axial resolution issues are thoroughly investigated in microscopic imaging applications, however, our discussion in the original submission aimed to highlight that the axial resolutions of early SIDH systems that were employed for capturing macroscopic scenes are significantly low. However, we realize that our claim was based on the

subjective assessment rather than quantitative analysis, mainly regarding the cited work [Kim, M. K. Full color natural light holographic camera. *Opt. Express* 21 (8), 9636–9642 (2013)], which seems to have very low axial resolution on the order of meters. Therefore, we decide to remove our claim about the axial resolution due to the lack of rigorous assessment of previous works and to focus on describing the low FoV issue of the systems. We thank you the reviewer for pointing out the unclear claim.

4. Once again reference 19 for geometric phase lens is not original. There are numerous articles on Geometric phase lens. What about the Nat. Photon article on CINCH where the birefringent lens was used to demonstrate FINCH? The seminal work in Nat. Photon 2008 on FINCH was also not referred. It can be seen that the random multiplexing as shown in the above mentioned article, still works for sparse objects. BTW the geometric phase lens is not a good direction to proceed especially due to the low beam overlap condition as clearly explained in the many articles on FINCH and SIDH. It has advantages but many disadvantages also. The shortened optical path can be achieved using dual lens FINCH with no loss. The GP approach achieves this but with loss – reduced overlap. I suggest the authors to do a thorough literature review and add relevant important articles. Many relevant articles by Nomura, Nobukawa, Tahara, Matoba and Juodkazis have not been discussed. After that description, the authors followed “However, poor image quality has remained a key unresolved issue in these studies” Can you explain the origin of this issue? Poor quality? Why? How? Unresolved? How? FINCH and SIDH are well-established techniques and there is no unresolved issue. For every problem, a solution has been found and published as articles. The only remaining problem in FINCH and SIDH is the need for multiple camera shots. Solutions are available with a choice of imaging characteristics to sacrifice – field of view, snr etc. The high FoV of GP needs to be explained.

We thank the reviewer for the constructive suggestion. We cited the Nature Photonics articles on CINCH and FINCH in the revised manuscript. As we discussed in the beginning of our response letter, we optimize the configuration of the incoherent holographic camera for capturing life-sized objects by enlarging the FoV. The primary way to achieve the large FoV in GP-SIDH systems is to exploit the partial beam overlap condition as shown in Fig. R1**b** in contrast to the perfect beam overlap condition that is typically employed in FINCH systems as shown in Fig. R1**a**. The FoV of both configurations is determined as follows:

$$FoV = 2 \operatorname{atan} \left(\frac{r_h}{z_h} \right) \quad (R1)$$

where r_h is the hologram radius and z_h is the distance between wavefront division device and image sensor. We can see that the partial beam overlap condition in the GP-SIDH systems provides the reduced z_h and increased r_h , resulting in the expanded FoV. As pointed out by the reviewer, the partial beam overlap is not an ideal condition because it leads to the degradation of the lateral resolution [Rosen, Joseph, Nisan Siegel, and Gary Brooker. Theoretical and experimental

demonstration of resolution beyond the Rayleigh limit by FINCH fluorescence microscopic imaging. *Optics express* 19 (27), 26249-26268 (2011)]. However, considering that our main goal is to capture life-size objects and that there is an inevitable trade-off between FoV and lateral resolution, we decide to increase the FoV at the expense of the lateral resolution.

Fig. R1 Comparisons of beam overlap conditions. **a** Perfect beam overlap condition provides high lateral resolution but reduced FoV. **b** Partial beam overlap expands FoV at the expense of lateral resolution. Gray areas indicate the beam overlap regions.

Although it looks as if the expansion of FoV can be simply achieved by placing the image sensor closer to the wavefront division device, the actual modification from the system in Fig. R1a to the system in Fig. R1b can be made only if two important conditions are satisfied:

Condition 1: Reducing z_h and increasing r_h are physically plausible.

Condition 2: The captured holograms should provide enough lateral and axial resolution. Otherwise, there is no benefit of using incoherent holographic cameras over conventional 2D cameras.

We found that the GP lens plays an important role in fulfilling those conditions. Regarding Condition 1, the GP lens can easily satisfy this condition: (1) the GP lens works with the transmission geometry unlike the LCoS SLM, which typically works with the reflection geometry, therefore we can reduce z_h down to a few millimeters, and (2) GP can be fabricated large enough so that the aperture size of the GP lens does not limit the hologram size r_h .

The validation of Condition 2 requires more careful consideration of the focal power of the wavefront division devices. In the following, we show that the positive and negative focal lengths of the GP lens is a key property to achieve a reasonable lateral and axial resolution for the system configuration in Fig. R1b. In our answer to Question 9, we derive the resolutions of the system as follows:

$$R_{Lateral} = 0.61 \frac{\lambda}{NA \cdot M_T}, R_{Axial} = \frac{2\lambda}{NA^2 \cdot M_A} \quad (R2)$$

$$R_{Lateral} = 0.61 \frac{\lambda z_r}{r_h} \cdot \frac{z_s}{z_h}, R_{Axial} = \frac{\lambda z_r^2}{r_h^2} \cdot \frac{(f_1 - f_2) z_s^3}{z_h (z_h f_2 z_s + f_1 (z_s z_h - 2 f_2 (z_h + z_s)))} \quad (R3)$$

where NA is the numerical aperture of system; M_T is transverse magnification; M_A is axial magnification; z_s denotes the distance between the imaging object and wavefront division device; f_1 and f_2 are two focal lengths generated by the wavefront division device; and z_r is the reconstruction distance of the hologram and given as [Katz, Barak, and Joseph Rosen. Super-resolution in incoherent optical imaging using synthetic aperture with Fresnel elements. *Optics express* 18(2), 962-972 (2010)]:

$$z_r = \frac{(f_1 z_s - z_h z_s + f_1 z_h)(f_2 z_s - z_h z_s + f_2 z_h)}{z_s^2 (f_1 - f_2)}. \quad (R4)$$

r_h is typically limited by the sensor size (refer to Question 9), therefore z_r is the key factor that determines the lateral and axial resolution of the system. As z_r depends on the two focal lengths f_1 and f_2 , we examine the best resolution condition based on three representative types of focal length pairs induced by existing wavefront division devices as shown in Fig. R2a-c. The first type (Fig. R2a) represents birefringence lenses [Siegel, Nisan, et al. High-magnification super-resolution FINCH microscopy using birefringent crystal lens interferometers. *Nature photonics* 10 (12), 802-808 (2016)] and the optical power difference between f_1 and f_2 are typically within 10%. The second type (Fig. R2b) corresponds to LCoS SLMs [Rosen, Joseph, and Gary Brooker. Digital spatially incoherent Fresnel holography. *Optics letters* 32 (8), 912-914 (2007)] and they have low f-numbers due to the small diffraction angle and aperture size. The third type (Fig. R2c) represents the GP lens used in our GP-SIDH system. They produce the negative and positive focal lengths with the same magnitude.

Fig. R2 Types of focal length pairs and corresponding simulated holograms. a Focal lengths have the same signs. **b** Transmissive beam and converging beam modulated with a positive focal length are generated. **c** Positive and negative focal lengths with the same magnitude are generated. **d, e, f** Simulated holograms for focal length pairs in **a, b, c**, respectively.

As the parametric space can be huge in Eq. R4, we fix z_h to 8 mm, which is a reasonable setting to achieve the large FoV in the configuration presented Fig. R1**b**. We also set the object position z_s to 39 cm, which is the center position of our target depth range [30 cm, 48 cm] of DeepIHC. Under these conditions, we calculate the lateral and axial resolutions based on the following focal length settings:

Table R1. Tentative lateral and axial resolutions provided by various wavefront division devices.

Focus type	Wavefront division device	Setting values	z_r (mm)	Lateral resolution (mm)	Axial resolution (mm)
$+f_1, +f_2$	Birefringence lens	1100 mm, 900 mm	5074	22.8	4304.2
$\infty, +f$	LCoS SLM	∞ , 1000mm	1033	4.6	839.6
$-f, +f$	GP lens (Ours)	-1000 mm, 1000 mm	520	2.5	444.1

Here, the focal lengths of the birefringence lens and LCoS SLM are set to have similar focal power with respect the GP lens. Although the exact values of the resolutions can vary depending on the focal length setting, Table R1 indicates that GP lens is a good direction to achieve the increased lateral and axial resolution when the system is optimized to have the large FoV. We also visually

examine the example holograms for three cases in Fig. R2d-f. The holograms show that higher spatial frequencies can be captured using the GP lens compared to the cases when birefringence lens or LCoS SLMs is used, indicating that higher lateral and axial resolutions can be obtained with the GP lens. We added the discussion on the large FoV of GP-SIDH to the supplementary information.

The advantage of using the GP lens for expanding FoV can be also observed in Table R2, which shows the summary of the FoVs presented in selected SIDH/FINCH systems. The wavefront divisions using LCoS SLMs [Kashter et al. (2015)] or the combination of a beam splitter and spherical mirror [Kim et al. (2013)] do not fulfill Condition 1 because the reflection geometry of system configuration inevitably introduces the large gap between the wavefront division device and image sensor. Birefringence lenses do not have such a limitation; therefore, the system presented in [Nobukawa et al. (2022)] can be potentially reconfigured to have large FoV. However, such a system modification would result in the loss in the lateral and axial resolutions. Therefore, we can see that the GP lens is the key component in expanding the FoV while maintaining the reasonable lateral and axial resolutions, as demonstrated in our GP-SIDH system or Tahara et al. (2021).

Table R2. FoVs of selected FINCH/SIDH systems.

	Wavefront division device	Phase modulation device	Condition 1	Condition 2	FoV
Our GP-SIDH system	Single GP lens	GP lens + polarized image sensor	O	O	36°
Tahara et al. (2021)	Double GP lens	GP lens + polarized image sensor	O	O	14°
Kim (2013)	Beam Splitter + spherical mirror	Piezo mirror	X	X	3°
Kashter et al. (2015)	Phase-only SLM (Sparse acquisition)	Phase-only SLM	X	X	2.14°
Nobukawa et al. (2022)	Birefringence lens	Checkerboard pattern DOE	O	X	2.5°

References for Table R2

Tahara, Tatsuki, and Ryutaro Oi. Palm-sized single-shot phase-shifting incoherent digital holography system. *OSA Continuum* 4 (8), 2372-2380 (2021).

Kim, M. K. Full color natural light holographic camera. *Optics Express* 21 (8), 9636-9642 (2013).

Kashter, Yuval, et al. Sparse synthetic aperture with Fresnel elements (S-SAFE) using digital incoherent holograms. *Optics express* 23 (16), 20941-20960 (2015).

Nobukawa, Teruyoshi, et al. Grating-based in-line geometric-phase-shifting incoherent digital holographic system toward 3D videography. *Optics Express* 30 (15), 27825-27840 (2022).

Regarding the origin of the poor image quality issue, we believe this originate from our system design choice of optimizing FoV and the characteristics of GP lens. As discussed earlier, the lateral resolution decreases due to the partial beam overlap setting in our system. Furthermore, the spatial variance of impulse response functions and optical imperfections of the GP lens additionally introduce the image quality degradation. As this image quality issue is specific to the SIDH systems that are targeting for capturing life-sized objects, we clarify the origin of the image quality problem in the revised manuscript. Please refer to line 79 – 119 in the revised manuscript.

In the revised manuscript, we also added the following references while discussing the GP lens-based SIDH systems and the single-shot capture systems.

Tahara, Tatsuki, and Ryutaro Oi. Palm-sized single-shot phase-shifting incoherent digital holography system. *OSA Continuum* 4 (8), 2372-2380 (2021).

Tahara, Tatsuki, et al. Single-shot phase-shifting incoherent digital holography. *Journal of Optics* 19 (6), 065705 (2017).

Nobukawa, Teruyoshi, et al. Single-shot phase-shifting incoherent digital holography with multiplexed checkerboard phase gratings. *Optics Letters* 43 (8), 1698-1701 (2018).

Vijayakumar, Anand, et al. Fresnel incoherent correlation holography with single camera shot. *Opto-Electronic Advances* 3 (8), 08200004 (2020).

5. In the manuscript, it is repeatedly mentioned – poor image quality which is not true. Yes in the 2013 article. But not in 2022. The current results of FINCH and SIDH are significantly better now and even commercialized by celloptic. Imaging results are available in their site.

We agree with the reviewer that FINCH or SIDH systems do not have any image quality issues when they are used as microscopic imaging systems, as clearly demonstrated by Celloptic, but we believe the image quality is still a significant issue when SIDH is employed as a general-purpose

camera. Apart from our GP-SIDH system, another good example can be found in a recent work [Tahara, Tatsuki, and Ryutarō Oi. Palm-sized single-shot phase-shifting incoherent digital holography system. *OSA Continuum* 4 (8), 2372-2380 (2021)]. Fig. R3 shows the example images obtained using GP-SIDH, the system proposed in Tahara et al, and the CINCH system developed by Celloptic. Our GP-SIDH system and the system developed by Tahara et al. are designed for imaging macroscopic objects, and they exhibit notably poor image quality compared to the images provided by Celloptic.

Fig. R3 Comparison of reconstructed images in FINCH/SIDH systems. **a** Target object in our GP-SIDH system and **b** reconstructed image from the raw hologram. **c** Target object in Tahara et al. and **d** reconstructed image from the raw hologram. **e, f** Representative images acquired using the CINCH system developed by Celloptic, Inc.

As addressed in our answer to Question 2, we believe this difference in image quality between microscopic imaging systems and the general-purpose camera comes from two factors: (1) we sacrifice the lateral resolution to achieve the higher FoV, and (2) the use of passive optical elements such as GP-lens in GP-SIDH or the system by Tahara et al. eliminate a room for active correction of aberrations. To our best knowledge, there is no SIHD system that can achieve both high lateral resolution and FoV, therefore we chose the FoV as our main target parameter of the system.

6. Again it is mentioned "As the neural network handles single-shot holograms, we also eliminate the need for multishot measurement, which has been typically practiced to reduce noise, hindering real-time applications" multishot measurement has been already eliminated by many methods. This is not eliminated by the authors by Deep learning for the first time. Even without deep learning, it

is possible to achieve single shot in FINCH and SIDH. Please do a thorough literature survey and add relevant articles on FINCH and SIDH.

As the reviewer correctly pointed out, several previous works capture the holograms with a single shot [Tahara et al. (2017), Nobukawa et al. (2018), Vijayakumar et al. (2020), Siegel et al (2021)]. However, to our best knowledge, denoising with a single shot capture has been first demonstrated by our work. The denoising methods proposed by previous works share the common approach of using multi-shot measurements [Katz et al. (2010), Bianco et al. (2018), Nobukawa et al. (2019)]. Even in single-shot capture systems, we can still obtain a notable improvement in image quality by temporally averaging multishot measurements. To avoid the ambiguity of our statement about the need for multishot measurement, we clarify that we can avoid the need for multishot measurement for denoising purposes in the revised manuscript, citing relevant references.

Line 132-136: *"As the neural network handles single-shot holograms, multishot measurements for denoising via temporal averaging are not necessary. It should be noted that despite the development of single-shot capture systems [30–33], denoising has typically been performed via multishot measurements in SIDH systems [34–36]."*

Tahara, Tatsuki, et al. Single-shot phase-shifting incoherent digital holography. *Journal of Optics* 19 (6), 065705 (2017).

Nobukawa, T.; Muroi, T.; Katano, Y.; Kinoshita, N.; Ishii, N. Single-shot phase-shifting incoherent digital holography with multiplexed checkerboard phase gratings. *Opt. Lett.* 43, 1698-1701 (2018).

Vijayakumar, Anand, et al. Fresnel incoherent correlation holography with single camera shot. *Opto-Electronic Advances* 3 (8), 08200004 (2020).

Siegel, Nisan, and Gary Brooker. Single shot holographic super-resolution microscopy. *Optics Express* 29 (11), 15953-15968 (2021).

Katz, Barak, Dov Wulich, and Joseph Rosen. Optimal noise suppression in Fresnel incoherent correlation holography (FINCH) configured for maximum imaging resolution. *Applied optics* 49 (30) 5757-5763 (2010).

Bianco, Vittorio, et al. Strategies for reducing speckle noise in digital holography. *Light: Science & Applications* 7(1). 1-16 (2018).

Nobukawa, Teruyoshi, et al. Sampling requirements and adaptive spatial averaging for incoherent digital holography. *Optics Express* 27 (23) 33634-33651 (2019).

7. The raw hologram pictures shown in Fig. 1b do not appear as FINCH/SIDH holograms. Can you show the recording plane in figure? Why for raw hologram and filtered hologram two images are shown? Why both have the image of the object 1 and not the object 2. Was a color camera used for recording? In FINCH and SIDH during reconstruction, the colour information is reconstructed at different locations due to the chromatic aberration introduced by diffractive components. How this problem was handled in this case? color based propagation still results in some error.

We thank the reviewer for the suggestion. We added the hologram data in all cases in the revised manuscript except for the holographic streaming demonstration in Fig. 6, whose main purpose is to demonstrate the real-time processing capability of DeepIHC. In Fig. 1 of the original manuscript, we presented the propagated holograms instead of the raw SIDH holograms. To show the quality and characteristics of the captured hologram data, we added the initial raw holograms as well in all cases. Regarding the confusing relationship between the object 1 and 2, we show two separate examples of capturing macroscopic objects in Fig. 1: the bear doll and the human face. In the revised manuscript, we included the detailed hologram data only for the bear doll and added the hologram data of the human face to the supplementary information. The recording plane of GP-SIDH is the image sensor plane and we specify it in Fig. S1 in the revised Supplementary Information.

Fig. R4 Revised Fig. 1 in the manuscript.

Fig. R5 Revised Fig. S1 in Supplementary Information.

Regarding the chromatic aberration introduced by the GP lens, we train the network in such a way that the output holograms have a clear focus and the lowest training loss at the object depth when propagated using the *conventional* ASM. This setting ensures that the output hologram is already compensated for the chromatic aberration because otherwise, the hologram propagated with the conventional ASM does not have a clear focus.

8. Please provide the images of hologram for all cases. The visibility of the holograms decrease with increase in spectral width. This is an important challenge. How this was rectified in your approach?

As suggested by the reviewer, we added the holograms for all cases in the revised manuscript. Since we aim to use GP-SIDH as a general-purpose camera, we do not constrain the characteristics of the light sources because such modifications would allow very limited environment settings for capturing scenes. The only spectral filtering system in our system is the color filter of the camera, and we attached the spectral width of the filter from the camera vendor.

Fig. R6 Spectral filter data. Adopted from <https://thinklucid.com/ko/product/phoenix-5-0-mp-polarized-model/>

As the reviewer pointed out, limiting the spectral width would increase the visibility of the holograms, however, it would also decrease the light collection efficiency. Given that we have both advantages and disadvantages with controlling the spectral width, we decide to simply rely on the default setting of the color filter of the polarized sensor.

9. Please explain the resolution limit - lateral and axial as per the GP overlap conditions.

As suggested by the reviewer, we theoretically derive the lateral and axial resolutions, and then we also experimentally confirm them based on our system parameter in the following response. We added the following results to the revised supplementary information.

1) Partial beam overlap condition of the GP-SIDH system

Before we derive the lateral and axial resolutions per the GP overlap conditions, we briefly show that the GP-SIDH system cannot be configured to have the perfect beam overlap condition. It should not impact our system design because we intentionally use the partial beam overlap condition to increase the FoV.

Fig. R7 Image formation in the GP-SIDH system.

Fig. R7 shows the image formation diagram of the GP-SIDH system. Here, z_s denotes the distance between the source object and the GP lens; r_{GP} represents the radius of the holograms; and z_p and z_n denote the locations of the virtual images formed by the positive and negative focal lengths of the GP lens, respectively. To satisfy the perfect overlap condition as in FINCH, the propagation angle θ_p induced by the positive focal length should be greater than propagation angle θ_n generated by the negative focal length:

$$\theta_p - \theta_n > 0, \quad (R5)$$

where

$$\theta_p = \text{atan}(r_{GP}/|z_p|), \theta_n = \text{atan}(r_{GP}/|z_n|). \quad (R6)$$

Therefore, we can check the following relationship instead:

$$\frac{r_{GP}}{|z_p|} - \frac{r_{GP}}{|z_n|} > 0 \quad (R7)$$

According to the lens formula, we have

$$z_p = f_{GP}z_s/(z_s - f_{GP}), z_n = -f_{GP}z_s/(z_s + f_{GP}). \quad (R8)$$

Then, we obtain the following relationship:

$$\frac{r_{GP}}{|z_p|} - \frac{r_{GP}}{|z_n|} = \frac{r_{GP}}{f_{GP}z_s} (|z_s - f_{GP}| - |z_s + f_{GP}|). \quad (R9)$$

Since both z_s and f_{GP} are positive values, $r_{GP}/|z_p| - r_{GP}/|z_n| < 0$ and thus $\theta_p < \theta_n$. Therefore, the perfect overlap condition cannot be met with the GP-SIDH systems. In the following sections, we consider only the partial overlap condition in our GP-SIDH system.

2) Theoretical derivation of lateral and axial resolutions of the GP-SIDH system

Fig. R8 System configuration of the GP-SIDH system.

In the following, we theoretically derive the lateral and axial resolution in the GP-SIDH system. The representative system configuration is shown in Fig. R8. In our GP-SIDH system, $f_{GP} = 1000 \text{ mm}$ and the target depth range is $z_s \in [300 \text{ mm}, 480 \text{ mm}]$; therefore, we consider the case when $f_{GP} > z_s$. In conventional digital holography, the effective lateral and axial resolutions are given as follows [Rosen, Joseph, Nisan Siegel, and Gary Brooker. Theoretical and experimental demonstration of resolution beyond the Rayleigh limit by FINCH fluorescence microscopic imaging. *Optics express* 19 (27), 26249-26268 (2011)]:

$$R_{lateral} = 0.61 \frac{\lambda}{NA \cdot M_T} \quad (R10)$$

$$R_{axial} = \frac{2\lambda}{NA^2 M_A} \quad (R11)$$

where M_T is the transverse magnification; M_A is the axial magnification, and NA is the numerical aperture (NA) of the system. The NA of the captured hologram is

$$NA = \frac{r_h}{z_r}, \quad (R12)$$

where r_h is the hologram radius and z_r is the reconstruction distance (not shown in Fig. R8) which is given as

$$z_r = \frac{(f_{GP}(z_s+z_h))^2 - (z_s z_h)^2}{2f_{GP} z_s^2}. \quad (R13)$$

Here, z_h denotes the distance between the GP lens and the image sensor. Therefore, the detailed

form of the resolutions can be expressed as

$$R_{lateral} = 0.61 \frac{\lambda}{NA \cdot M_T} = 0.61 \frac{\lambda z_r}{r_h} \cdot \frac{z_s}{z_h} \quad (R14)$$

$$R_{axial} = \frac{2\lambda}{NA^2 M_A} = \frac{\lambda z_r^2}{r_h^2} \cdot \frac{z_s^3}{z_h f_{GP}(z_s + z_h)} \quad (R15)$$

As the beam overlap is proportional to z_h , we study the dependence of the lateral and axial resolutions on z_h . We set z_s to 390 mm, which is the center distance of the depth range of DeepIHC. Then, the corresponding reconstruction depth z_r becomes 530 mm. In order to fully determine the axial and lateral resolutions, we need to compute the hologram radius r_h , which can be determined using the following criterion:

$$r_h = \min(r_{OPL}, r_{Nyquist}, r_{CMOS}) \quad (R16)$$

Here, r_{OPL} , $r_{Nyquist}$, r_{CMOS} represent the maximum hologram radius limited by optical path difference, sensor sampling frequency, and size of CMOS sensor, respectively.

Firstly, r_{OPL} is determined by the interference formation condition as shown in Fig. R8:

$$OPD = \Delta OPL = |OPL_{pos} - OPL_{neg}| < coherence\ length = \frac{\lambda^2}{\Delta\lambda} \quad (R17)$$

For a given r_{OPL} , the corresponding hologram radius r_p and r_n at the GP lens are given as

$$r_p = \frac{z_p}{z_p + z_h} r_{OPL}, r_n = \frac{z_n}{z_n + z_h} r_{OPL}, \quad (R18)$$

where z_p and z_n denote the locations of the virtual images formed by the positive and negative focal lengths of the GP lens, respectively.

$$z_p = \frac{f_{GP} z_s}{z_s - f_{GP}}, z_n = -\frac{f_{GP} z_s}{z_s + f_{GP}}. \quad (R19)$$

Therefore, the optical path difference can be derived using a simple geometric consideration:

$$OPD = \left\{ (z_s^2 + r_p^2)^{\frac{1}{2}} + (z_h^2 + (r_{OPL} - r_p)^2)^{\frac{1}{2}} \right\} - \left\{ (z_s^2 + r_n^2)^{\frac{1}{2}} + (z_h^2 + (r_{OPL} - r_n)^2)^{\frac{1}{2}} \right\} \leq \frac{\lambda}{\Delta\lambda} \quad (R20)$$

r_{OPL} is the maximum value that satisfies the condition in Equation R20. Assuming the illumination wavelength of 550 nm, the spectral width of 100 nm of our system provides r_{OPL} of 65 mm. However, this derivation assumes that r_p and r_s are not limited by the aperture size of the GP lens. In case the GP lens is a limiting factor, r_{OPL} is computed using the following formula:

$$r_{OPL} = r_{GP} \cdot \frac{f_{GP} z_s + f_{GP} z_h - z_h z_s}{f_{GP} z_s} \quad (R21)$$

The GP lens used in our system has a 2-inch diameter; therefore, the aperture size of the GP lens

is the limiting factor of r_{OPL} and the final value is 26 mm.

Secondly, $r_{Nyquist}$ describes the limitation posed by the sampling rate of the image sensor, which can be derived from the following relationship between the pixel pitch Δx of the image sensor and central wavelength λ :

$$r_{nyquist} = z_r \frac{\lambda}{\Delta x}. \quad (R22)$$

The center wavelength of 550 nm and the pixel pitch 3.45 μm in our systems provides $r_{Nyquist}$ of 40 mm.

Lastly, $r_{CMOS} = 7$ mm in our GP-SIDH, and the final hologram radius is determined as

$$r_h = \min(r_{OPL}, r_{Nyquist}, r_{CMOS}) = \min(26 \text{ mm}, 40 \text{ mm}, 7 \text{ mm}) = 7 \text{ mm} \quad (R23)$$

Therefore, the hologram radius is currently limited by the size of the image sensor. This indicates that the GP-SIDH system has room for enhancing lateral and axial resolutions by employing the synthetic aperture strategy, as mentioned in the manuscript.

Fig. R9 Dependence of the lateral and axial resolutions on z_h a Effective lateral resolution and FoV as a function of z_h . b Effective axial resolution and FoV as a function of z_h .

Using the computed hologram radius, we obtain the effective lateral and axial resolutions as a function of z_h in Fig. R9. As the distance z_h between the GP lens and image sensor increases, the beam overlap also increases. The result shows that as the beam overlap increases, the lateral and axial resolutions increase as well. However, as discussed in our answer to Question 4, the increased z_h leads to the reduced FoV. We use $\mu\text{p/mm}$ unit to show the opposite trends of the resolutions

and FoV. Therefore, we use z_h of 8 mm (indicated with the black dashed line) in our system as a compromise between the FoV and resolutions. For this setting, the lateral and axial resolutions are calculated as 2.5 mm and 444 μ m in our system. In the following section, we also experimentally confirm the computed resolution values. We found that the axial resolution seems to be large, however, our analysis shows that large axial resolution *does not* imply that objects that are separated by a distance smaller than the axial resolution cannot be differentiated. We can still observe clear defocus effect within the depth range that is smaller than the axial resolution; therefore, we discuss the implication of the axial resolution in the context of 3D imaging.

3) Experimental confirmation of lateral and axial resolutions of the GP-SIDH system

a) Lateral resolution

To validate the theoretical value of the lateral resolution, we conducted two experiments using two point-sources and resolution targets.

Fig. R10 Lateral intensity profiles of the two point sources for various lateral separations. **a, b, c** Reconstructed images of the two point sources when the separation was set to 0 mm, 2 mm, and 3 mm, respectively, and **d, e, f** corresponding intensity profiles along the x-axis. **g** Reconstructed image of the hologram that captures the USAF resolution target. The minimum resolvable pattern is indicated by a white rectangle.

The theoretical derivation on the lateral resolution is based on the point spread functions of the point light sources. To experimentally reproduce this setting, we generate two point sources and set the lateral position difference using a beam splitter. We set three different lateral separation values of 0 mm, 2mm, and 3 mm and reconstruct the images and corresponding intensity profiles for each case, as shown in Fig. R10. We found that the lateral separation of 2 mm exhibits the

intensity profile (Fig. R10e) similar to the Rayleigh criterion, and the lateral separation of 3 mm (Fig. R10f) indicates that two point sources are already well resolvable. Therefore, we can conclude that the Rayleigh resolution is approximately 2 mm, which well matches the theoretical value of 2.5 mm. We also confirmed the lateral resolution by imaging the USAF resolution target, as shown in Fig. R10g. The minimum resolvable pattern indicated by the white rectangle was 0.353 lp/mm, which is converted to a lateral resolution of 2 mm.

b) Axial resolution

We measure the axial resolution based on the full width at half maximum (FWHM) of the axial profiles of point light sources, following the approach in [Siegel, Nisan, and Gary Brooker. Improved axial resolution of FINCH fluorescence microscopy when combined with spinning disk confocal microscopy. *Optics express* 22 (19) 22298-22307 (2014)]. We captured the hologram of a point source placed at $z_s = 390$ mm and reconstructed the axial intensity profiles (orange solid line) as a function of z_r , as shown in Fig. R11a. We also simulated a hologram for the point source placed at the same position and acquired an almost identical axial intensity profile (blue solid line) in Fig. R11a. We express both intensity profiles as a function of z_s in Fig. R11b. The FWHM of the experimental and simulated results are 428 mm and 482 mm, respectively. Therefore, we see that the validation results of the experiment and simulation match reasonably well with the theoretical axial resolution of 444 mm.

Fig. R11 Axial intensity profiles of a point source. **a** Axial intensity profiles of the simulated and measured holograms of a point source. The profiles are derived as a function of the reconstruction distance z_r . **b** Axial intensity profiles expressed in terms of the physical distance z_s .

c) Implication of axial resolution in the context of 3D camera

Fig. R12 Reconstructed images from the captured holograms at various depths. DeepIHC holograms are acquired for the target at 300 mm (Top), 390 mm (Middle), and 480 mm (Bottom) and images are reconstructed from the holograms at three different reconstruction depths at 300 mm (Left), 390 mm (Center), and 480 mm (Right).

Although we confirmed the good match between the theoretical predictions and experimental results, we found that the axial resolution of 444 mm seems to be large compared to the depth range of the DeepIHC which is 180 mm. We note that the axial resolution of the system does not fully describe the important aspect of the 3D camera system: the amount of the defocus blur. When we use the GP-SIDH system for daily-use cameras and show acquired holograms on holographic displays, the essential role of the captured holograms is to provide the visually noticeable defocus

effect rather than to enable the quantification of the exact axial separation between the objects. We believe our system provides such a defocus capability as demonstrated in Fig. R12. We placed the same image at three different depths, namely, 300 mm, 390 mm, and 480 mm, and acquired the corresponding holograms using DeepIHC. The top, middle, and bottom rows present reconstructed images from the DeepIHC holograms of the target at 300 mm, 390 mm, and 480 mm, respectively. Although the depth separation between targets is well below the axial resolution, we can still clearly see the defocus effect and easily pinpoint the best-focused plane.

We also examine the simulated axial intensity profiles of the point sources placed between 300 mm and 480 mm with a separation of 30 mm, as shown in Fig. R13. This depth configuration matches the seven equally-spaced depth planes used in our training dataset capture. We see that the amount of separation is well below the FWHM, however, we can expect that the defocus blur is still observed within the depth range of 180 mm.

Fig. R13 Simulated axial intensity profiles of the point sources placed with a physical separation of 30 mm.

We found that the axial resolution is not an ideal measure of the 3D camera; therefore, proper quantification of depth resolution should be investigated. This will require the consideration of the amount of required blur for daily-use 3D cameras and an understanding of the depth perception of the camera or display users. We believe studying such issues is beyond the scope of our paper, and we leave further investigation as future work.

10. The image of the letters are ok but the image is noisy. Please provide holograms for all cases.

The images of the letters are clear because they are artificially created and overlaid on the captured hologram. To make the procedure clear, we added the illustration of the AR processing and the full hologram data for each intermediate step in the revised Fig. 7.

Fig. R14 Illustration of hologram editing process added to the revised Fig. 7.

11. The limits on the AI method needs to be addressed.

We thank you the reviewer for the suggestion. In the original submission, we discussed the susceptibility to the lighting conditions, limited depth range, and outlook on processing high-resolution holograms with AI methods. By incorporating the concerns raised by the reviewers, we additionally discuss the shortcoming of using simple propagation kernels and the limitation of using pseudo 3D dataset in the revised manuscript.

Line 432-439: *“Extending the depth range requires a neural network with a larger receptive field; therefore, large propagation kernels [47] beyond those standard convolution layers used in DeepLHC should be investigated. We also found that the diverse depth configurations in real-world scenes are challenging to incorporate during neural network training due to the difficulty of extracting precise depth information in arbitrary 3D scenes. Therefore, a new strategy for collecting fully 3D real-world datasets with appropriate RGB-D reference data should be devised.”*

Minor

1. There is a typo in incoherent in Fig. 1

We thank the reviewer for pointing out the type. We fixed it in the revised figure.

Reviewer #2 (Remarks to the Author):

The submitted manuscript, "Towards holographic streaming systems: from real-time acquisition of the real world to holographic display", introduces a learned algorithmic approach for a holographic camera system. Specifically, the holographic camera operates with incoherent light, and the learned algorithmic approach helps retain visual quality by denoising.

A conclusion of my review could be found at the end of this text.

We thank the reviewer for reviewing our work and suggesting valuable comments.

Typos, Style, Grammar

1. Please consult with a native speaker. In the title, you may need to be saying: "from the real-time acquisition of real-world ".

We thank the reviewer for pointing out the grammatical error. Our previous submission was proofread by Nature Research Editing Service, but we regret that some corrections were not perfect. To ensure better readability, we consulted again with the proofreading service for the revised manuscript.

- Line 91-94, your line 87 already identifies the poor image quality problem.

We thank the reviewer for bringing our attention to this point. We added more details about the poor image quality problem to address Reviewer 1's concerns and removed the repeated mention about the image quality problem in line 91-94 in the revised manuscript.

- I treat figures as mini-pages. Please check with the common culture in the journal you are submitting for. I trust that you may need to write acronyms in full at first appearance in a figure's

caption. Otherwise, readers must visit various places in the manuscript to understand each term. It degrades readability, in my opinion.

We thank the reviewer for the constructive advice. As suggested by the reviewer, we rewrote acronyms in full in the revised manuscript.

Comments

- Line 50, none of the works in 2 or 4-6 deals with narrow eye box issues, but they aim for visual quality. Consider citing: Kuo, Grace, Laura Waller, Ren Ng, and Andrew Maimone. "High resolution étendue expansion for holographic displays." *ACM Transactions on Graphics (TOG)* 39, no. 4 (2020): 66-1.

The reference [4] (Yu et al. Ultrahigh-definition dynamic 3d holographic display by active control of volume speckle field. *Nature Photonics* 11(3), 186-192 (2017)) deals with the narrow viewing angle issue of desktop holographic displays, which translates into narrow eye box issues in near-eye holographic displays, and is one of the foundational works of the suggested paper. Since the suggested paper directly addresses the narrow eye box issue in the context of near-eye displays, we agree with the reviewer that it is a relevant paper to our discussion, and we added the reference to the revised manuscript.

- Line 104, you may consider highlighting multishot as helping with temporal averaging of noise. I trust that temporal averaging is more common in computer-generated holography communities than multishot learning.

We thank the reviewer for pointing out the implication of multishot measurement. We agree with the reviewer that multishot measurements help with temporal averaging of noise. Given that our work focuses on the holographic camera system rather than the generation of computer-generated holograms, we also would like to clarify that multishot measurement is a commonly used term in the incoherent holographic imaging community due to the following reasons: (1) the phase-shifting holograms typically require multishot measurements to capture a single hologram and (2) several previous works reduces the noise via capturing multiple holograms [Katz et al. (2010), Bianco et al. (2018), Nobukawa et al. (2019)]. Our original intention was to emphasize that our work neither employs multishot measurements for constructing a single hologram nor requires multiple holograms for denoising. Our system captures a single hologram with a single measurement, and the neural network outputs the denoised hologram from a single hologram input. Since our claim was unclear as suggested by Reviewer 1 as well, we added the following discussion by addressing both reviewers' comments.

Line 132-136: *"As the neural network handles single-shot holograms, multishot measurements for denoising via temporal averaging are not necessary. It should be noted that despite the development of single-shot capture systems [30–33], denoising has typically been performed via multishot measurements in SIDH systems [34–36]."*

- In figure 1 and your introduction, I gather that your previous work is geometric-phase lens-based self-interference incoherent digital holography. Your contribution is to derive a learned hologram capture algorithm.

Yes, we also consider that developing the learned hologram capture algorithm is one of our main contributions.

- Line 122-124, the statement is not clear and came out of the blue. What is the problem with the spatial variance of the impulse response function?

We thank the reviewer for bringing up the rightful concern about the sudden introduction of the technical problem without discussing enough context. In the supplementary information of the previous submission, we showed several exemplary holograms which indicate that the impulse response functions are spatially invariant and also scene-dependent, therefore it is not straightforward to apply simple aberration correction. Furthermore, accurate aberration correction typically requires spatial- and depth-dependent consideration, however, we do not have any prior information about 3D geometry of the object, which adds another difficulty to our problem setting. As these challenges are the main motivation for the development of our algorithm, we added a brief summary of this observation to the main manuscript in the resubmission.

Line 156-167: *"The degraded image quality can be attributed to the hologram formation model. Captured incoherent holograms can be described as incoherent summations of impulse response functions from individual points in the captured 3D object. However, the impulse response functions in GP-SIDH systems have spatial and depth dependencies, which are highly challenging to characterize experimentally (see Supplementary Information Section 1.2). Moreover, even if we can acquire this information, 3D information about the target 3D object is required in the inverse correction of the optical aberrations, and this information is difficult to obtain. In addition to the image degradation introduced by the spatial variance in the impulse response functions, we observe that the signal-to-noise ratio (SNR) significantly decreases as the scene complexity increases (see Supplementary Information Section 1.2)."*

- Line 125, if I gather correctly, as the number of points to resolve in a scene increases, the captured data gets highly convoluted. Thus, you can no longer accurately pinpoint a reliable solution with a forward and inverse model. Sounds fair.

As the reviewer pointed out correctly, the difficulty of modeling a forward and inverse propagation is the main motivation for employing the deep learning-based techniques.

- Line 129-131, would it be fair to call your learned method a denoiser? It looks like you already agree in lines 146-147.

We believe it is fair to call our method a denoiser. We do not augment images or enhance the resolution but remove the noise, which is the primary way to enhance the visual quality of the holograms captured by our system.

- In line 166, in my view, you need to dedicate a paragraph on how your camera hardware works actually. Because your readers are left with a learning routine and capture that they will have a hard time following.

We thank the reviewer for the constructive suggestion. We added the description of the characteristics of captured holograms and how to process them on the abstract level. The detailed theory based on equations still remain in the supplementary material in order to keep the main manuscript as concise as possible.

Line 148-156: *"The GP-SIDH, in which the recording plane matches the sensor plane, is used to capture a raw hologram as shown in Fig. 1b. To reconstruct the object image, we propagate the raw hologram to the object plane using the angular spectrum method, as shown in Fig. 1c, and compute the intensity image, as shown in Fig. 1e. The extremely poor image quality of the reconstructed image suggests that the raw hologram (Fig. 1b) captured with the GP-SIDH system alone cannot provide practically usable 3D data (see Supplementary Information Section 1 for more details on this system)."*

- Line 166-170, this part needs a little bit of decyphering. You capture a hologram that represents a volume bounded by propagation distances ranging between 30 to 48 cms. Your hologram captures are 1024 by 1024 pixels. Your hologram represents the propagation of d_i , but in code, you propagate it to the central plane d_c using the Angular Spectrum Method.

As the reviewer pointed out correctly, we propagate the captured hologram to the central plane at d_c regardless of the object depths. The main reason for choosing this strategy is that the depth

information of the captured object is not available during the validation. During the training process, we intentionally place the planar objects at specific depths because we need to make sure that the denoised holograms have a clear focus at the corresponding object depths. However, during the validation or in practical capture scenarios, the objects can be placed at any depth or they can have multi-depth geometry, therefore, we believe it is reasonable to assume that the depth information of the target object is unknown. For this reason, we decided to set the common propagation depth d_c for any input holograms. We added the implication of the setting the common propagation depth to the revised manuscript.

Line 217-220: *"It should be noted that the hologram is propagated to the central plane regardless of the object depth. We chose this strategy because we cannot access the depth information of the captured objects during the validation stage."*

- Line 170. Is there a color-independent angular spectrum method? I always thought light propagation kernel in any approach is wavelength dependent, no?

As the reviewer mentioned, it is correct that the kernels are wavelength dependent in any approach, but *the propagation depth is independent* of the wavelengths in traditional ASM. Our intention was to highlight that *the propagation depths are dependent* on the wavelengths due to the optical aberrations of the GP lens. We found that the "color-independent angular-spectrum method" is a confusing term; thus, we changed the term to "depth-corrected angular spectrum method" in the revised manuscript.

- Line 171-172. Yes, when you propagate to large distances, the kernel size of your light propagation simulation would be larger. Yes, this would translate to a need for larger receptive fields in a learned algorithm. If I follow correctly, you suggest that having the hologram at the center of the volume minimizes the requirement in large kernel sizes. Let me highlight that there are also large kernel-based learned algorithms that do not follow classical five-by-five or 11 by 11 convolutional layers: Kavakli, Koray, Hakan Urey, and Kaan Akşit. "Learned holographic light transport." Applied Optics 61.5 (2022): B50-B55.

We thank the reviewer for introducing an interesting paper. Employing small kernels as in our network would prohibit us from extending the depth range. Following the suggestion by the reviewer, we added related discussion to the revised manuscript and mentioned the paper as one of the future research directions.

Line 432-435: *"Extending the depth range requires a neural network with a larger receptive field;*

therefore, large propagation kernels [47] beyond those standard convolution layers used in DeepIHC should be investigated.

- Line 177, are these boundary artifacts related to aliasing?

These boundary artifacts are related to the beam diffraction rather than aliasing. During the ASM propagation, the diffracted beam from a single pixel diverges toward the nearby pixels, and the coverage is proportional to the diffraction angle θ and the propagation distance.

$$A = 2d \tan \theta \quad (\text{R24})$$

In our system, the maximum coverage corresponds to 30 pixels and our margin is set to 120 pixels. This margin seems to be excessively large, however, we also aim to demonstrate that our system can handle 720 x 720 resolution in real-time given that most high-quality modern media files typically support at least 720p resolution. We added the related discussion to the manuscript.

Line 226-232: *“the sufficient margin of 120 pixels, which is set to prevent boundary artifacts that may occur during ASM propagation when diffracted beams diverge and contribute to nearby pixels. For the propagation within the [30 cm, 48 cm] range, the maximum expansion corresponds to 30 pixels in our system; however, we set the margin rather aggressively to show that DeepIHC can handle a resolution of 720 × 720 in real time, as most high-quality media files support at least 720p resolution.”*

- Line 186, which perceptual loss are you using, though? [29] explains several, no?

We thank the reviewer for pointing out the lack of the detailed description of the perceptual loss. We used the perceptual loss obtained using the output of the 7-th convolution layer of the VGG-19 network [Simonyan and Zisserman, Very deep convolutional networks for large-scale image recognition (2014)]. We added the exact form of the perceptual loss and detailed setting in the revised manuscript along with the citation of Simonyan et al..

Line 241-244: *“where $\varphi_{j,k}$ is the feature map obtained by the k -th convolution layer before the j -th maxpooling layer in the VGG-19 network [44]. $W_{j,k}$ and $H_{j,k}$ denote the dimensions of the feature maps. We use the activation from the VGG_{3,3} convolutional layer.”*

- Line 186, I strongly suggest the authors release a public code repository as their work gets accepted.

While we strongly believe that releasing codes public should be a widely adopted culture and we have put significant efforts to resolve administrative concerns, unfortunately we found that there are several internal issues hindering us from releasing the public code repository especially given that most of the authors are affiliated to the private company Samsung. Our company is conservative about releasing the public code especially when the core work is patented. We hope the reviewer would understand our situation.

- Line 186-187, where do we get the ground truth images for the images captured in holograms? Is there a secondary camera in your system that captures conventional photographs of a scene, and if so, how do you deal with the transformations required between a holographic camera and a conventional camera. Later, I read section 3 of your supplementary, and I gather that you are using a display as a proxy and mapping what is on the screen to ground the truth image. This is fair, but help me, please also understand, wouldn't that mean you are dealing with a two-dimensional hologram dataset but not three-dimensional captures? How does your network generalize to the case of three-dimensionality in this case? Or is this system fixed focus? Do you still need to adjust the focus manually, as in the supplementary video?

We believe the best way to describe our situation is that the captured *holograms* are three-dimensional, whereas captured *objects* for the training dataset are pseudo 3D: planar objects placed at various depths.

Holographic cameras do not have an intrinsic fixed focus because the captured holograms can be freely propagated to an arbitrarily chosen depth. For example, Supplementary Video Part 4 shows simulated perceived images at three different focal depths reconstructed from the *single* hologram. If our system had a fixed focus and therefore we had to adjust the focus to get each focal image manually, it would have been impossible to capture the exact same facial expressions and movements in these three reconstruction images shown in Supplementary Video Part 4. This suggests that holograms captured by our system carry full 3D information.

Although DeepIHC can capture 3D holograms, the target objects in our training dataset are not fully 3D. Over the course of our research, we considered two possible options for acquiring the fully 3D dataset, but we found those options extremely challenging or limited.

Option 1) We capture various real objects and extract the corresponding ground-truth images and depth maps using existing algorithms.

(Challenge) It is extremely challenging to prepare real objects with diverse shapes and textures. Also

acquisition of ground-truth RGB-D images for training is highly challenging.

Option 2) We display virtual 3D objects on a coherent holographic display system and capture them by DeepIHC. We use the RGB-D information used for generating virtual 3D objects as the ground-truth data.

(Limitation) This strategy contradicts the assumption that our camera is working with incoherent lights. Also, the image quality, color gamut, and FoV will be highly limited by the specification of the holographic display used.

For those reasons, we developed the strategy to use the pseudo-3D training dataset despite its limitations. To avoid the limited depth handling as much as possible, we put the target object at various depths, and we also simulate the multi-depth holograms by artificially synthesizing captured holograms. Fig. 5 indicates that DeepIHC can still work on multi-depth objects even though we did not use any training hologram that captured multi-depth objects. We discuss the use of pseudo 3D dataset in the revised manuscript along with the difficulty of collecting fully 3D dataset.

Line 204-208: *"It should be noted that employing holographic displays to generate reference 3D images is not a viable option because most holographic displays operate under coherent illumination conditions, which contradicts the working principle of incoherent holographic cameras."*

Line 435-439: *"We also found that the diverse depth configurations in real-world scenes are challenging to incorporate during neural network training due to the difficulty of extracting precise depth information in arbitrary 3D scenes. Therefore, a new strategy for collecting fully 3D real-world datasets with appropriate RGB-D reference data should be devised."*

- Line 190, how many holograms do you have in total in your train set? Will you be able to release this set in the future publicly? Do you have large epochs because you have a small dataset?

We used 5250 holograms in total. As some of the authors are affiliated to private company, we regret that it is not feasible to release the training dataset publicly due to the company's policy. We stopped the training when the validation loss becomes saturated, and we observed a tendency that a larger dataset requires fewer epochs and results in lower validation loss. We did not experiment with dataset with a larger number of holograms as it already takes a day to capture the dataset and five days to train them with the current dataset. We added information about the training to the method section in the revised manuscript.

- In lines 211-212, convolutional layers tend to "hallucinate" things. When you test your network against a test validation, were there cases where you have an image with a feature that has non-existing in the training dataset? Say if, in your train dataset, there is no a boat or a chair, etc., would it still denoise properly? I encourage the authors to add such a case to their comparison in figure 3.

As the captured holograms carry diffraction patterns rather than exact image features and the main role of the neural network is removing the noise rather than generating images, we expect the network not to heavily rely on the image features for denoising. To confirm this, as suggested by the reviewer, we examined the following images as shown in Fig. R15. We found that the objects similar to the letters 'POLICE', the ancient statue, the red car, and the yellow helmet are not shown in the images in the training dataset (DIV2K dataset, image numbers 1–750), but we can see that they provide the average enhancement of PSNR of 10.6 dB. Due to the inclusion of hologram data in all cases we now have an excessive number of figures in the main manuscript, therefore we added the material below to the supplementary information.

Fig. R15 Validation of denoising images with unseen objects. **a** Validation target images that contain objects which did not appear in the training dataset. **b** Images reconstructed at the corresponding target object depths from the raw holograms. **c** Images reconstructed from the filtered holograms.

- Across the figures, besides Figure 2, you don't show a hologram you captured, right? So when you say "raw hologram", you mean "**classical reconstruction** from a raw hologram", am I correct?

As the reviewer pointed out, it is more accurate to say "classical reconstruction from a raw hologram". We fixed the labels in the revised manuscript.

- Line 269, what does this holographic display entail?

Up to Line 269 (Section. 2.3), we demonstrated the development of the denoising algorithm for incoherent holographic cameras. Although our learning-based camera system alone can be used to collect 3D contents of the real-world and reconstructed images can be displayed on conventional 2D displays, we believe that the best platform to display the captured holograms with a full-3D effect is a holographic display. Therefore, we built the holographic stream system consisting of our camera capture system and holographic display to demonstrate that the real-time acquisition and display of real-world holograms are achievable. Given that most of holographic display research demonstrate their capability by displaying CGHs computed from computer graphics scenes or preprocessed RGB-D data of the real-world, we believe that the real-time capture and display of holograms of the real-world is one of the key achievements of our work.

I am unaware of a similar complete holographic display system in the literature. Thus, I believe the provided system is novel in that respect. On the other hand, if I gather correctly, the primary contribution here is the learned algorithmic approach, a series of convolutional layers followed by activation blocks. Furthermore, this specific network operates only in the hologram domain; both input and output are holograms. Previous works cited in their literature consist of networks that only work on the hologram domain (hologram in, hologram out; here [5] U-Net takes a hologram and spits out a hologram). I value this research and understand the importance of the research work. However, I need to be convinced. I trust the authors can help me by carefully addressing my comments in their responses. As is, I am leaning towards rejection.

We appreciate the reviewer for acknowledging the novelty and importance of our work. Although our work is inspired by seminal work [5] (Peng, Yifan, et al. Neural holography with camera-in-the-loop training. *ACM Transactions on Graphics (TOG)* 39 (6), 1-14 (2020)) in a way that hardware calibration can be realized by the deep neural network, we believe our work can be notably differentiated from their work. We found that however, stating that we are the first to work on a hologram-in-hologram-out neural network can be overclaiming, therefore we specify that our network is designed for incoherent holographic cameras in the revised manuscript.

Line 194-197: *"To the best of our knowledge, our denoising algorithm is the first neural network proposed for denoising incoherent holograms that are acquired by incoherent holographic cameras."*

Fig. R16 Relationship among key components of various hologram generation/acquisition techniques.

Fig. R16 summarizes the key components of various hologram generation/acquisition techniques. Conventional computer-generated holograms are typically generated based on intermediate representation of 3D computer graphics scenes or 3D real-world scenes assuming aberration-free ideal holographic displays (the essential components are linked by black solid lines). Recent learned algorithmic approaches [Peng et al. (2020), Chakravarthula et al. (2020)] provide exceptionally high-quality images by considering the optical imperfections of actual holographic display devices, but they still rely on the intermediate representations of 3D scenes (major components linked by green solid lines). Although our work is also developed based on deep learning techniques, we tackle a distinctive problem from previous works: we aim to acquire the holograms of the real-world in real-time using incoherent holographic cameras (components linked by orange solid lines) rather than to generate holograms from the intermediate representations. Therefore, our technique is not simply an advanced solution that improves the image quality of existing approaches, but we enable a missing part of holographic ecosystem that has not been realized before: the direct acquisition of real-world scenes in hologram format and restoring the image quality in real time so that the captured real-world holograms can be instantly displayed on holographic displays. We believe the development of the DeepIHC system for this application and its demonstrations are the most important novelty and contribution to the holographic camera and display community.

As discussed above, our work and learned hologram generation methods have different target domains and optimization goals. Therefore, the hardware system configuration, overall hologram processing pipeline, preparation of dataset, and neural network training procedures are notably different. Furthermore, the learned hologram generation approaches can be additionally considered for DeepIHC systems because we denoise holograms assuming that they will be displayed on ideal holographic displays rather than holographic displays with optical imperfections. However, the

existing learned hologram generation approaches would not be immediately applicable to our system because they rely on the RGB-D data of the target scenes assuming that the depth information is already known, whereas depth information is implicitly encoded in incoherent holograms. We added this discussion in the revised manuscript as well.

Peng, Yifan, et al. Neural holography with camera-in-the-loop training. *ACM Transactions on Graphics (TOG)* 39 (6), 1-14 (2020)

Chakravarthula, Praneeth, et al. Learned hardware-in-the-loop phase retrieval for holographic near-eye displays. *ACM Transactions on Graphics (TOG)* 39 (6), 1-18 (2020).

Line 453-463: *“Although our work is inspired by the recent development of learned hologram generation methods [5, 6], we did not consider optimizing the filtered holograms for specific holographic displays and instead focused on the different goal; the acquisition of high-quality holograms of real-world scenes. In future works, it would be interesting to explore how to optimize the holograms output by DeepIHC for actual holographic displays. This research direction poses a new challenge because the basic assumption of the learned hologram generation method, namely, that the depth information is already known, does not hold for incoherent holographic cameras, as the depth information is implicitly encoded in incoherent holograms.”*

REVIEWERS' COMMENTS

Reviewer #1 (Remarks to the Author):

The authors have suitably revised the manuscript by including supporting information obtained from a thorough analysis of the proposed incoherent holography system. The manuscript is much better now. I believe that this work will start a new promising direction for FINCH. I do not have any further concerns.

Reviewer #2 (Remarks to the Author):

The authors have addressed my comments. However, I also see that the other reviewer has a standing question. At the time, my suggestion leans toward acceptance.

point-by-point response to the reviewers' comments

REVIEWERS' COMMENTS

Reviewer #1 (Remarks to the Author):

The authors have suitably revised the manuscript by including supporting information obtained from a thorough analysis of the proposed incoherent holography system. The manuscript is much better now. I believe that this work will start a new promising direction for FINCH. I do not have any further concerns.

Reviewer #2 (Remarks to the Author):

The authors have addressed my comments. However, I also see that the other reviewer has a standing question. At the time, my suggestion leans toward acceptance.

=> We appreciate Reviewer #1 and #2 for their helpful comments. We could increase the quality of our manuscript thanks to the helpful advice of both Reviewers.